# Synergism between CMG helicase and leading strand DNA polymerase at replication fork

Zhichun Xu[1,6], Jianrong Feng[2,6], Daqi Yu[2,6], Yunjing Huo [1,6], Xiaohui Ma[2,6], Wai Hei Lam[1], Zheng Liu[3], Xiang David Li [3], Toyotaka Ishibashi[2], Shangyu Dang [2,4,5] ✉ & Yuanliang Zhai [1] ✉

The replisome that replicates the eukaryotic genome consists of at least three engines: the Cdc45-MCM-GINS (CMG) helicase that separates duplex DNA at the replication fork and two DNA polymerases, one on each strand, that replicate the unwound DNA. Here, we determined a series of cryo-electron microscopy structures of a yeast replisome comprising CMG, leading-strand polymerase Polε and three accessory factors on a forked DNA. In these structures, Polε engages or disengages with the motor domains of the CMG by occupying two alternative positions, which closely correlate with the rotational movement of the single-stranded DNA around the MCM pore. During this process, the polymerase remains stably coupled to the helicase using Psf1 as a hinge. This synergism is modulated by a concerted rearrangement of ATPase sites to drive DNA translocation. The Polε-MCM coupling is not only required for CMG formation to initiate DNA replication but also facilitates the leading-strand DNA synthesis mediated by Polε. Our study elucidates a mechanism intrinsic to the replisome that coordinates the activities of CMG and Polε to negotiate any roadblocks, DNA damage, and epigenetic marks encountered during translocation along replication forks.

In eukaryotes, DNA replication is tightly regulated to ensure a faithful duplication of genomic DNA[1–3]. The replisome, which is composed of a CMG (Cdc45-Mcm2-7-GINS) helicase and at least two DNA polymerases at its core, is responsible for catalyzing DNA replication[4–7]. Additional accessory factors are also required for an efficient replication of chromatin DNA with high fidelity[2,8–10].

The assembly of replisomes starts with the association of origin recognition complex (ORC) with replication origins in the early G1 phase[2,11]. ORC then serves as a platform to recruit two copies of Mcm2-7 complexes onto origin DNA as a head-to-head double hexamer (DH) encircling double-stranded DNA (dsDNA)[11–16]. Upon entering S phase, multiple origin-firing factors work in concert to convert the inert Mcm2-7 DH into two active replisomes[17–20]. First, Dbf4-dependent kinase Cdc7 (DDK) phosphorylates Mcm2/4/6 subunits to assemble Sld3-7 and Cdc45 onto the Mcm2-7 DH[21–26]. Next, S-cyclin-dependent kinase (S-CDK) phosphorylates Sld3 and Sld2, enabling the recruitment of other firing factors such as Dpb11, Sld2, GINS, and DNA polymerase ε (Polε) to form the pre-initiation complex (pre-IC)[17–19,27]. This leads to the formation of two CMG complexes at the origin DNA[28,29]. Finally, Mcm10 activates the CMG helicases[30,31]. With the incorporation of other factors, such as replication protein A (RPA) and DNA polymerase α-primase, the bidirectional replisomes are established at the divergent replication forks and translocate along single-stranded DNA (ssDNA) to unwind duplex DNA for DNA synthesis. In addition to these

[1]School of Biological Sciences, The University of Hong Kong, Hong Kong, China. [2]Division of Life Science, The Hong Kong University of Science & Technology, Hong Kong, China. [3]Department of Chemistry, The University of Hong Kong, Hong Kong, China. [4]Southern Marine Science and Engineering Guangdong Laboratory (Guangzhou), Guangzhou, China. [5]HKUST-Shenzhen Research Institute, 518057 Nanshan, Shenzhen, China. [6]These authors contributed equally: Zhichun Xu, Jianrong Feng, Daqi Yu, Yunjing Huo, Xiaohui Ma. ✉e-mail: sdang@ust.hk; zhai@hku.hk

factors, a multitude of accessory factors, such as Ctf4, Csm3, Tof1, Mrc1 and FACT (FAcilitates Chromatin Transcription), are required to enhance the rate of fork progression along chromatin DNA and maintain genome stability[2,5–9].

Significant progress has been made in recent years toward understanding the regulation of eukaryotic replisomes on two fronts. First, the in vitro reconstitution of the DNA synthesizing system using purified yeast proteins has identified a minimal set of proteins for in vitro replication of both naked and chromatinized DNA as templates[8,10,18,19,32]. The available in vitro systems also enabled single-molecule analysis to understand the dynamic properties of the replisome components on fork DNA during the process of DNA unwinding and synthesis[33–35]. Second, high-resolution cryo-EM structures of CMG helicases and CMG in complex with other replication factors have been determined[11,36–45]. The CMG structures reveal that Cdc45 and GINS are assembled onto the Mcm2-7 complex via the N-terminal domain (NTD) subdomain As (NTD-As) of Mcm2/3/5, subsequently stabilizing the NTD tier of MCM ring in a close and planar conformation. This stable NTD tier is then used by CMG at fork DNA as the front end to translocate along leading strand template 3′–5′ for DNA unwinding[42]. A nonsymmetric hand-over-hand rotary mechanism has been proposed as a potential translocation mechanism of *Drosophila* CMG helicase during DNA unwinding[37]. The roles of several CMG associating factors in regulating replisome activities are beginning to emerge. For example, the Ctf4 trimer (And-1 homolog in human) sits on top of the interface of Cdc45 and GINS, almost perpendicular to the NTD ring[43,46], serving as a hub to coordinate lagging strand DNA synthesis with other processes such as parental histone recycling and sister chromatid cohesion. Csm3 and Tof1, components of fork protection complex (FPC), associate with the N tier of CMG through binding to the NTD-As of Mcm2/6/4/7 to engage with parental duplex DNA[41]. The binding of Tof1-Csm3 to CMG has been shown to enhance the rate of fork progression[9]. In addition, the cryo-EM structure of the yeast CMG bound with Polε (referred to as CMGE) was previously reported[38]. In this CMGE structure, Polε binds to CMG at two major components: the motor domains of the MCM ring and the Psf1 B1 domain of GINS. This special arrangement places Polε close to the potential path of leading strand DNA exiting CMG helicase. However, due to the limited resolution of the Polε-CMG interfaces (-7.5 Å) in the CMGE structure, the detailed interaction between these two engines was not fully resolved[38]. Recently, a cryo-EM structure of human CMG in complex with Polε, TIMELESS-TIPIN, CLASPIN and AND-1 was also determined[44]. The overall architecture of this human replisome is very similar to its yeast counterpart[38,41]. The structure of Polε holoenzyme shows that Dpb3 and Dpb4 insert into the wedge between the catalytic (cat) NTD and non-cat CTD of Pol2 to regulate the stability of this catalytic NTD module[40]. Together, these biochemical and structural studies have significantly advanced our understanding of the regulation of the eukaryotic replisome at a molecular level.

To drive replication initiation and maintain the rate of fork progression, CMG helicase must be tightly coupled to Polε in order to coordinate DNA unwinding and synthesis in a translocating replisome[4]. However, how the activities of these two engines are synchronized at the replication fork during DNA unwinding remains largely unknown. In this study, we determined a series of cryo-EM structures of a yeast leading strand replisome, which provide unprecedented snapshots of the working mechanism of CMG helicase in motion featuring a coupled Polε cycling on and off the MCM ring at the replication fork along with DNA translocation.

## Results

### Overall structure of a yeast leading strand replisome

DNA replication is a highly dynamic process in which DNA unwinding by replicative helicase is in locked step with DNA polymerases to regulate fork progression. To understand how CMG helicase is coordinated with Polε during DNA unwinding, we assembled a leading-strand replisome complex by first incubating CMG helicase with forked DNA in the presence of a slowly hydrolyzable ATP analog ATP-γ-S, followed by the addition of Polε, Ctf4, Tof1 and Csm3 (Supplementary Fig. 1a). The reaction mixture was then subjected to glycerol gradient fixation. The corresponding fractions containing successfully assembled replisome were further processed for cryo-EM studies. Good particles sorted by several rounds of 2D and 3D classification were used for the final 3D reconstruction. A cryo-EM density map for a replisome was generated at an overall resolution of 3.2 Å comprising all included components at a preformed forked DNA (Supplementary Fig. 1f, Supplementary Table 1).

In this replisome structure, the CMG helicase serves as a core to assemble Tof1-Csm3 onto the NTD face of the MCM ring at the Mcm6/4/7 side and the Ctf4 homotrimer onto the interface of Cdc45 and GINS while Polε associates with both GINS and the CTD face of the MCM ring at the Mcm2/5/3 side (Supplementary Fig. 2a–e). Parental duplex DNA protruding from the NTD tier of the MCM ring is captured by Tof1-Csm3 and bent toward the Mcm6/4/7 side (Supplementary Fig. 2a, c, d, f), resembling previously reported structures of the yeast CMG bound with Tof1-Csm3[41] and the in vitro assembled human replisome[44]. Interestingly, when Tof1 and Csm3 are absent, the parental duplex DNA becomes highly flexible (Supplementary Figs. 1j and 2g–l), suggesting an important role of Tof1-Csm3 in positioning duplex DNA for unwinding. In this complex, only the DNA fork junction inserted into the NTD tier of the MCM ring is visible (Supplementary Fig. 2l). This property of Tof1-Csm3 in positioning and stabilizing the parental duplex DNA explains its role in enhancing the rate of fork progression by promoting efficient DNA unwinding[9].

The lagging strand DNA is highly dynamic and was not resolved in our structure. However, cryo-EM densities corresponding to the leading strand DNA are observed inside the MCM central channel (Supplementary Fig. 2f). A stable, helical ssDNA fragment is attached to the inner surface of the MCM-CTD ring on the Mcm2/5/3 side (Supplementary Fig. 2e, f). Notably, Polε also binds to the outer surface of the same CTDs in a position poised to capture the leading strand DNA exiting from the CMG helicase (Supplementary Fig. 2e, f). The architecture of Polε in the replisome is similar to that of the previous structure of the yeast CMGE[38] and the human replisome[44]. To better understand how these two engines are coupled in the replisome, we performed a local refinement focused on Polε to analyze the surrounding CMG regions using replisome particles with a stable Polε (Supplementary Fig. 1h). This analysis largely improved the local resolution of the relevant regions and generated a final density map of the replisome complex at 2.98 Å (Supplementary Fig. 1i). In this structure, most residues at the interfaces of Polε and CMG are clearly resolved, enabling unambiguous atomic modeling of the portions of Pol2 and Dpb2 involved in CMG docking.

### High-resolution details of Dbp2-NTD in coupling Polε to Psf1

In the replisome structure, Polε interacts with CMG through four docking sites (discussed under different sections) (Fig. 1a–d). At docking site (DS) 1, Dpb2-NTD (residues 12–98), which is highly flexible in the apo Polε holoenzyme[40], is stably anchored onto the B-domain of Psf1 (residues 161–208) with a buried surface area (BSA) of 1140 Å$^2$ (Fig. 1a, e–h). This interaction is known to be essential for incorporating Polε into the replisome in yeast[47]. The Dpb2-NTD forms a left-handed superhelical bundle consisting of four helices and three connecting loops (Fig. 1f, Supplementary Fig. 3). The α helices 1-2 (α1-2) and loops 2-3 (L2-3) of Dpb2-NTD are highly positively charged, matching a negatively charged surface on the Psf1-B domain (Supplementary Fig. 3b–d). This structural arrangement leads to a tightly coupled interface formed between Psf1 and Dpb2, which involves 7 hydrogen bonds, 1 salt bridge, and several van der Waals interactions

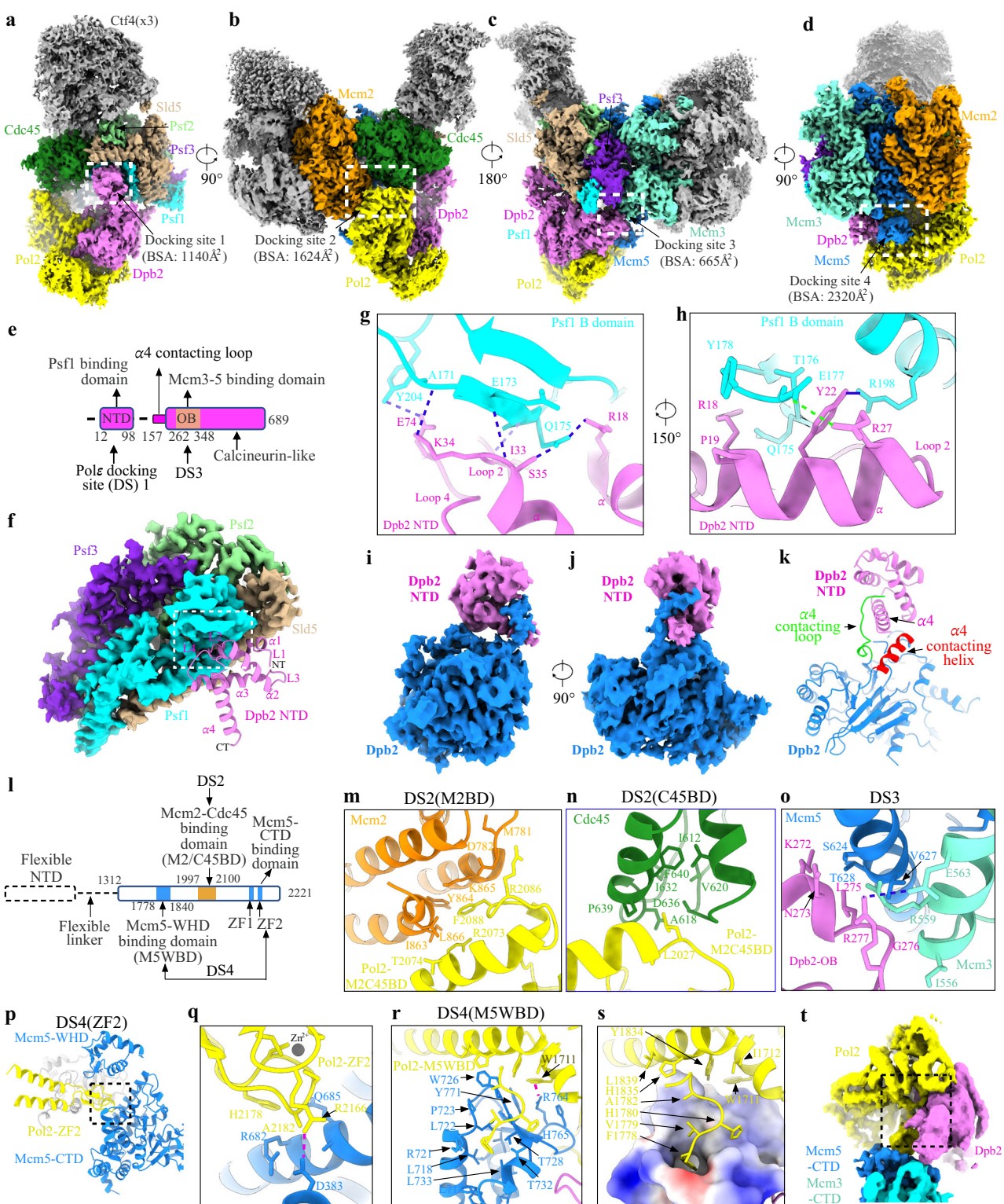

(Fig. 1g, h). We noted that Dpb2-NTD shares a low sequence similarity between yeast and higher eukaryotes. However, sequence comparison guided by the Dpb2 structure shows high conservation for this region across various species (Supplementary Fig. 3a). The structure of the NTD of POLE2 (human homolog of Dpb2) also contains a bundle of four helices arranged similarly to its yeast counterpart[44,48] (Supplementary Fig. 3e). Moreover, human POLE2-NTD and PSF1 exhibit similar patterns of polarized surface charge as their yeast counterparts (Supplementary Fig. 3g–i). These results suggest a conserved mechanism for incorporating the leading strand polymerase into the replisome via the interactions between Dpb2/POLE2 and Psf1 from yeast to human.

Dpb2 can be divided into a helical NTD, a highly disordered linker (residues 99–156), and an inactivated calcineurin-like phosphodiesterase (PDE) domain (residues 157–689) (Fig. 1e). The α4 helix of Dpb2-NTD is strategically sandwiched between a structured loop (residues 157–167) and another helix (residues 381–393) close to the PDE domain (Fig. 1i–k). This compacted structure suggests that Dpb2-NTD also contributes to stabilizing Polε on CMG by providing structural support to its PDE domain.

**Fig. 1 | Detailed interactions of Polε with CMG helicase. a–d** Side views of the cryo-EM density map of the replisome, showing the docking sites of Polε on CMG (dashed squares). For clarity, Mcm6/4/7 subunits and Tof1-Csm3 are removed in (**d**), highlighting the interaction of Polε with Mcm5. Relevant subunits are color-coded and labeled as indicated. Measured buried surface areas for each interface in (**a–d**) are labeled. BSA: buried surface area. **e** Schematic domain organization of Dpb2. **f** Interaction between Dpb2 and Psf1 with Dpb2-NTD shown in cartoon presentation and GINS displayed with its cryo-EM density map. The α helices and connecting loops of Dpb2-NTD are labeled as indicated. α: helix, L: loop; NT: N-terminus; CT: C-terminus. **g, h** Magnified views of the boxed region in (**f**), showing the detailed interaction between Dpb2-NTD and Psf1 B domain in cartoon presentation. The hydrogen bonds are indicated by dashed lines in blue; Salt bridge between E177 of Psf1 and R27 of Dpb2 is indicated by the dashed line in green. **i, j** Density map of Dpb2 showing the interaction between the NTD and PDE domain of Dpb2. **k** Atomic model of Dpb2 with α4 contacting loop and helix highlighted in green and red, respectively. **l** Schematic domain organization of Pol2. Dashed lines denote highly disordered segments that are not resolved in the replisome structure. **m** Magnified view of the boxed region in (**b**) highlighting the interaction details of Pol2 with the AAA+ domain of Mcm2 in cartoon presentation. **n** Magnified view of the boxed region in (**b**) illustrating the interaction of Pol2 with Cdc45. **o** Magnified view of the boxed region in (**c**) showing the Dpb2 OB domain contacting the CTD interface of Mcm3:5. Hydrogen bond between R277 of Dpb2 and E563 of Mcm3 is indicated by the blue dashed line. **p** Interaction between Pol2-ZF2 (yellow) and Mcm5-CTD (blue). **q** Magnified view of the boxed region in (**p**). The hydrogen bond between R2166 of Pol2 and D383 of Mcm5 is indicated by the red dashed line. $Zn^{2+}$ atom is shown as a gray sphere. **r** Pol2 (yellow) docking to Mcm5-WHD (blue). Cation-π interaction between R764 of Mcm5 and W1711 of Pol2 is indicated by the dashed line in red. **s** Same as (**r**) but shown with the electrostatic surface potential map of Mcm5-WHD, highlighting Mcm5-WHD binding to a region of Pol2 rich in hydrophobic residues. Positively charged residues are in blue; negatively charged residues in red; nonpolar and hydrophobic residues in white. **t** A replisome conformer exhibiting a highly flexible Mcm5-WHD.

## Flexible catalytic NTD of Pol2 in the leading strand replisome

In the Polε holoenzyme, the non-essential catalytic NTD of Pol2[49] adopts a relatively rigid conformation by interacting with the CTD and Mooring helix of Pol2 and Dpb3/4[40]. In contrast, this catalytic module appears highly dynamic in the leading strand replisome (Supplementary Figs. 1k and 4a–c). To understand if CMG binding results in a flexible Pol2-NTD, we compared the Polε structures from both the holoenzyme and the replisome. Using Dpb2-PDE as a reference, we observed an obvious rotational shift in the CTD of Pol2 toward the CMG in the replisome (Supplementary Fig. 4d–h). The displacements are as large as 6–10 Å in the regions of Pol2-CTD responsible for contacting Pol2-NTD and Dpb3/4 (Supplementary Fig. 4g). It is likely that these conformational changes weaken the interaction network of Pol2-NTD, Pol2-CTD and Dpb3/4 observed in the holoenzyme, giving rise to a highly dynamic NTD of Pol2 in the replisome. Our 2D class averages show diffuse signals of Pol2-NTD at locations surrounding the non-catalytic Pol2-CTD and regions as far as the MCM-CTD ring where leading strand DNA exits (Supplementary Fig. 4b, c). Furthermore, some relatively weak densities corresponding to Pol2-NTD are evident at various positions around Pol2-CTD in four replisome maps derived from 3D classification focused on the catalytic domain of Polε (Supplementary Fig. 1k). We speculate that this dynamic feature of Pol2-NTD on CMG provides the polymerase with a high degree of flexibility to handle DNA template under different physiological conditions.

## Polε docking sites on the MCM ring of CMG

In the replisome structure, the rigid non-catalytic module of Polε binds to the motor domains of the MCM ring at three sites (docking sites 2-4, Fig. 1b–d). At docking site 2 (DS2), one edge of Pol2-CTD (referred to as Mcm2 and Cdc45 binding domain, M2/C45BD) engages with the inter-subunit interface between Mcm2 and Cdc45 mainly through hydrophobic interactions (Fig. 1b, l–n). As a result, two flanking loops of Pol2-M2/C45BD, which are disordered in the holoenzyme, become structured and engage with an α helix associated with the ZF2 motif of Pol2 (Supplementary Fig. 4i). It is likely that these loops help stabilize the interaction between Pol2-M2/C45BD and CMG. Relative to the DS2, the docking site 3 (DS3) is located on the opposite side of the MCM ring, where the Dpb2-OB domain engages the CTD interface of Mcm3 and Mcm5 (Fig. 1c, o). This arrangement positions Pol2-ZF2 and the dead polymerase fold of Pol2-CTD on top of Mcm5-CTD at docking site 4 (DS4) (Fig. 1p–r), creating a hydrophobic binding surface (referred to as Mcm5-WHD binding domain, M5WBD) for Mcm5-WHD (Fig. 1d, r, s). Notably, Mcm5-WHD is dispensable for Polε docking onto the MCM ring. Our focused 3D classification identified 8 classes of replisome in which Mcm5-WHD exhibits different degrees of stability on Pol2 (Supplementary Fig. 1n). In one of these classes, Mcm5-WHD becomes highly flexible and completely invisible while Polε remains stably engaged with the MCM ring (Fig. 1t, Supplementary Fig. 1n). As shown in the apo CMG structure[39] and Mcm2-7 single hexamer[50], Mcm5-WHD resides in the MCM pore to occlude its central DNA binding channel. Our results suggest that Polε binding to the MCM ring provides an anchoring point for Mcm5-WHD on Pol2 to facilitate DNA threading by CMG helicase.

## The role of forked DNA in Polε binding to the motor domains of the MCM ring

The total BSA between the non-catalytic module of Polε and the MCM ring is 4609 Å² (Fig. 1b–d), which is significantly larger than the BSA of Dpb2-NTD binding to Psf1 B domain (1140 Å²) (Fig. 1a). These contact surfaces suggest that Polε forms a much stronger interaction with the MCM ring than with Psf1 B domain. It is likely that Polε could be incorporated into the replisome independent of Dpb2-NTD. To test this hypothesis, we purified a mutant Polε with the NTD (residues 1–90) of Dpb2 removed (referred to as PolεΔ2N) and examined its binding affinity to CMG (Fig. 2a, lanes 1-3). Our in vitro pulldown assays show that PolεΔ2N can still interact with the DNA-engaged CMG, although its binding affinity is lower than that of Polε-WT (Fig. 2a, lanes 6-7). This result indicates that the MCM ring mediates a direct binding of Polε to the CMG helicase, independent of Dpb2-NTD. Interestingly, in the absence of DNA, the recruitment of PolεΔ2N onto the CMG complex is largely inhibited (Fig. 2a, lane 11). In contrast, Polε-WT exhibits a strong binding affinity to the apo helicase (Fig. 2a, lane 10). These observations suggest that the MCM ring of the DNA-free CMG assumes conformation(s) incompetent for Polε docking. Under this situation, Dpb2-NTD becomes indispensable for coupling Polε to CMG via its interaction with Psf1. These results also highlight the pivotal role of forked DNA in regulating the incorporation of Polε into the leading strand replisome.

## Polε docking onto the MCM ring is essential for replication initiation

To investigate the role of Polε in binding to MCM ring, we constructed a series of *pol2* mutants with specific MCM binding domains deleted, either individually or in various combinations (Fig. 2b). To assess the impact of these mutations on cell viability, we transformed the relevant constructs for ectopic expression of the mutant Pol2 proteins under the native *POL2* promoter into a tight *pol2-iAID* (improved auxin inducible degron) strain that allowed conditional depletion of endogenous Pol2-AID fusion proteins[51]. Our findings showed that deletion of a single MCM binding domain (*pol2ΔM2BD*, *pol2ΔM5WBD*, and *pol2ΔZF2*), a complete DS (*pol2ΔDS2* and *pol2ΔDS4*), or even two binding domains (one each from DS2 and DS4, *pol2ΔM2BDΔM5WBD* and *pol2ΔM2BDΔZF2*) did not affect the growth of *pol2-iAID* cells under restrictive conditions (Fig. 2c). However, simultaneous removal of both DS2 and DS4 (*pol2ΔDS2+4*) resulted in inviable cells (Fig. 2c). To determine whether these deletion mutations disrupt the interaction

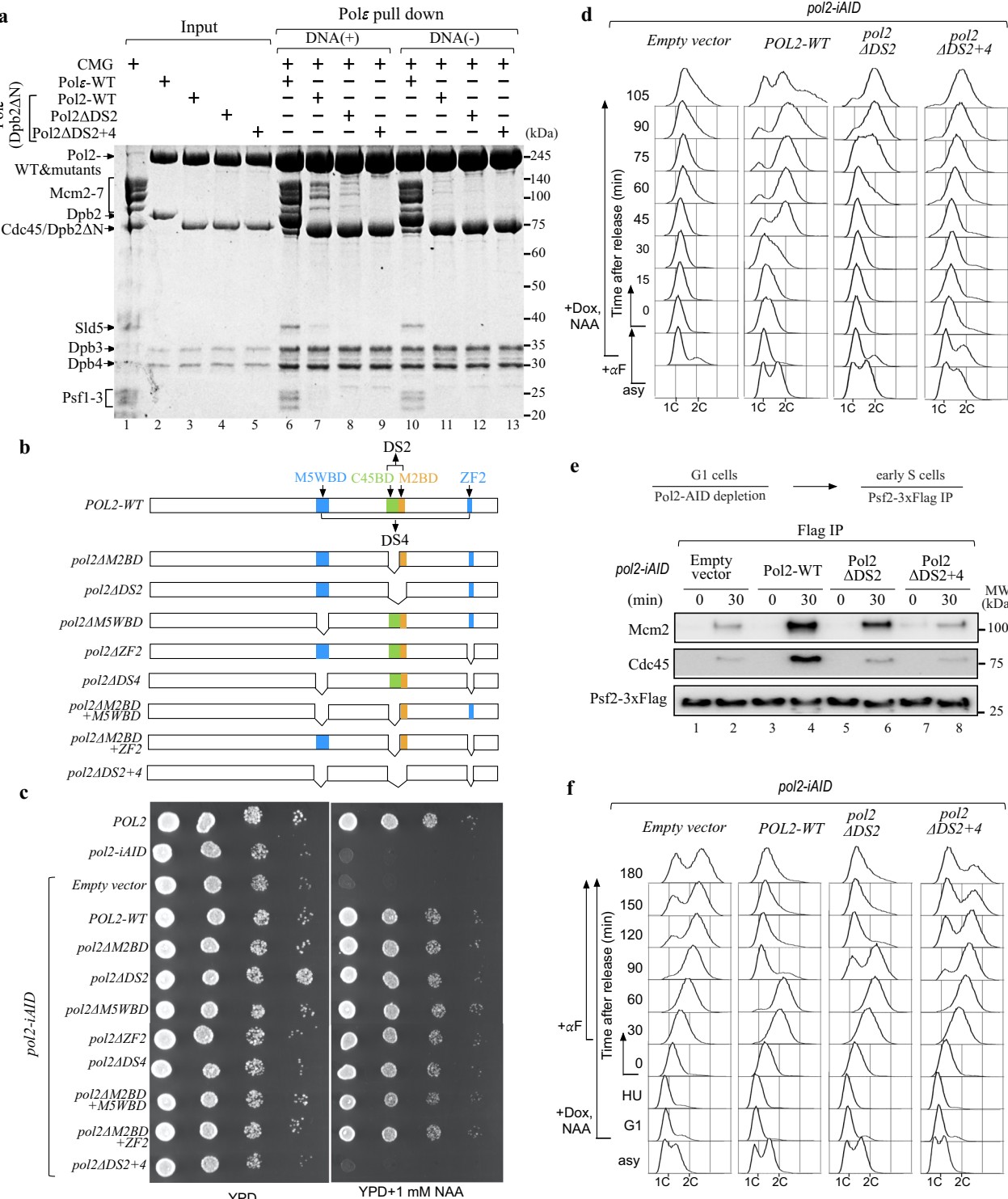

**Fig. 2 | The physiological function of Polε-MCM coupling in replication initiation and S phase progression. a** In vitro CMG pulldown assays to examine the binding affinity of Polε, PolεΔ2N, and the related mutants (Pol2ΔDS2 and Pol2ΔDS2+4) to CMG in the presence or absence of DNA. Similar results were obtained in two independent experiments. **b** Schematic illustration of *pol2* mutant construction. **c** Polε-MCM coupling is essential for cell viability. The indicated truncation mutants of Pol2 were tested for complementation of a depletion of Pol2-AID, the expression of which is under the control of Tet promoter. YPD: yeast extract-peptone-dextrose, NAA: 1-naphthalene acetic acid, Dox: doxycycline.

**d** Flow cytometry profiles of cells collected from the α factor block-and-release assays with the indicated *pol2-iAID* strains under restrictive conditions. **e** The lysates of the indicated samples were prepared for Flag IP with anti-FLAG beads. Each sample was analyzed with antibodies against Flag, Cdc45, and Mcm2. Similar results were obtained in two independent experiments. Source data are provided as a source data file. **f** Flow cytometry profiles of cells collected from the hydroxyurea (HU) block-and-release assays. α factor was added after cells were released from HU arrest.

between Polε and the MCM ring, we purified two Pol2 mutant-containing PolεΔ2N complexes (refer to as Pol2ΔDS2 and Pol2ΔDS2+4) for in vitro pulldown assays (Fig. 2a, lanes 4-5). Our results revealed that the affinity of PolεΔ2N binding to the DNA-engaged CMG complex was significantly reduced with Pol2ΔDS2 (Fig. 2a, lane 8) and almost completely abolished with Pol2ΔDS2+4 (Fig. 2a, lane 9), suggesting that both DS2 and DS4 are crucial for stabilizing the binding of Polε to the MCM ring.

To understand the role of the Polε-MCM coupling in vivo, we performed an α factor block-and-release assay in which the relevant *pol2-iAID* cells were pre-synchronized in the G1 phase with α factor to deplete Pol2-AID proteins before releasing them into fresh medium under the restrictive condition. Our FACS analysis showed that the *pol2ΔDS2* mutant displayed an obvious delay in S phase entry compared to the *POL2-WT* (Fig. 2d). This defect became more severe in the *pol2ΔDS2+4* mutant, which is similar to the control strain bearing an empty vector (Fig. 2d). To investigate whether the delayed S phase entry is due to impaired origin firing, we analyzed the firing reaction by immunoprecipitating Psf2-Flag in the *pol2* mutant and control cells after depleting endogenous Pol2-AID proteins. In the *POL2-WT* cells, Psf2-Flag co-precipitated with both Mcm2 and Cdc45 only with early S phase cells (30 min after release from G1 arrest) (Fig. 2e, lanes 3-4), indicating that the CMG complex was properly assembled. However, the levels of both Mcm2 and Cdc45 co-precipitated with Psf2-Flag were reduced for the *pol2ΔDS2* mutant (Fig. 2e, lanes 5-6) and largely suppressed for the *pol2ΔDS2+4* mutant (Fig. 2e, lanes 7-8) respectively, highlighting that the CMG formation was interrupted. These results demonstrate that a stable Polε-MCM coupling is essential for CMG formation to drive replication initiation.

Next, we sought to determine whether the interaction between Polε and the MCM ring is necessary during the elongation phase of DNA replication in vivo. To this end, the *pol2-iAID* strains bearing *pol2* mutants were first pre-synchronized in G1 with α factor and then released into a medium containing hydroxyurea (HU) under a permissive condition to allow activation of early origins. After the depletion of Pol2-AID, the mutant cells were released from HU into a fresh medium under restrictive conditions. We found that the majority of the *pol2ΔDS2* and *pol2ΔDS2+4* mutant cells replicated their DNA and entered the next cell cycle afterward, as evidenced by the appearance of a G1 population (Fig. 2f). These results suggest that the Polε-MCM coupling may be dispensable for DNA replication following origin firing in vivo. However, we cannot rule out the possibility that a residual amount of Pol2-AID is sufficient to sustain leading-strand DNA synthesis in the mutant cells. Notably, previous studies demonstrated that Polε function in leading-strand DNA synthesis can be substituted by DNA polymerase Polδ but with a lower extension rate[49,52,53].

## Polε cycling on and off MCM ring in relation to DNA translocation by CMG

While the binding of Polε to the MCM ring may not be essential for S phase progression, the activities of these two engines are likely coordinated in the replisome to facilitate helicase translocation and leading-strand DNA extension. It is believed that conformational changes in the motor domains of the MCM ring drive DNA translocation by CMG[36,39]. A recent structural study with DNA-engaged *Drosophila* CMG proposed a nonsymmetric hand-over-hand mechanism for the rotational movement of DNA around the MCM pore during DNA unwinding[37]. Given the fact that DNA plays a crucial role in mediating the Polε-MCM coupling (Fig. 2a), we hypothesized that the DNA movement within the CMG may be coupled to a periodic binding of Polε to the MCM ring. To test this hypothesis, we performed a rigorous 3D classification focused on the C tier of the MCM ring of all replisome particles (Supplementary Fig. 1l, m). This analysis identified six conformational states of the replisome, referred to as States I-VI, which vary in both Polε stability (Conformers I-IV, Fig. 3) and the location of leading-strand DNA lining the MCM pore (Fig. 4).

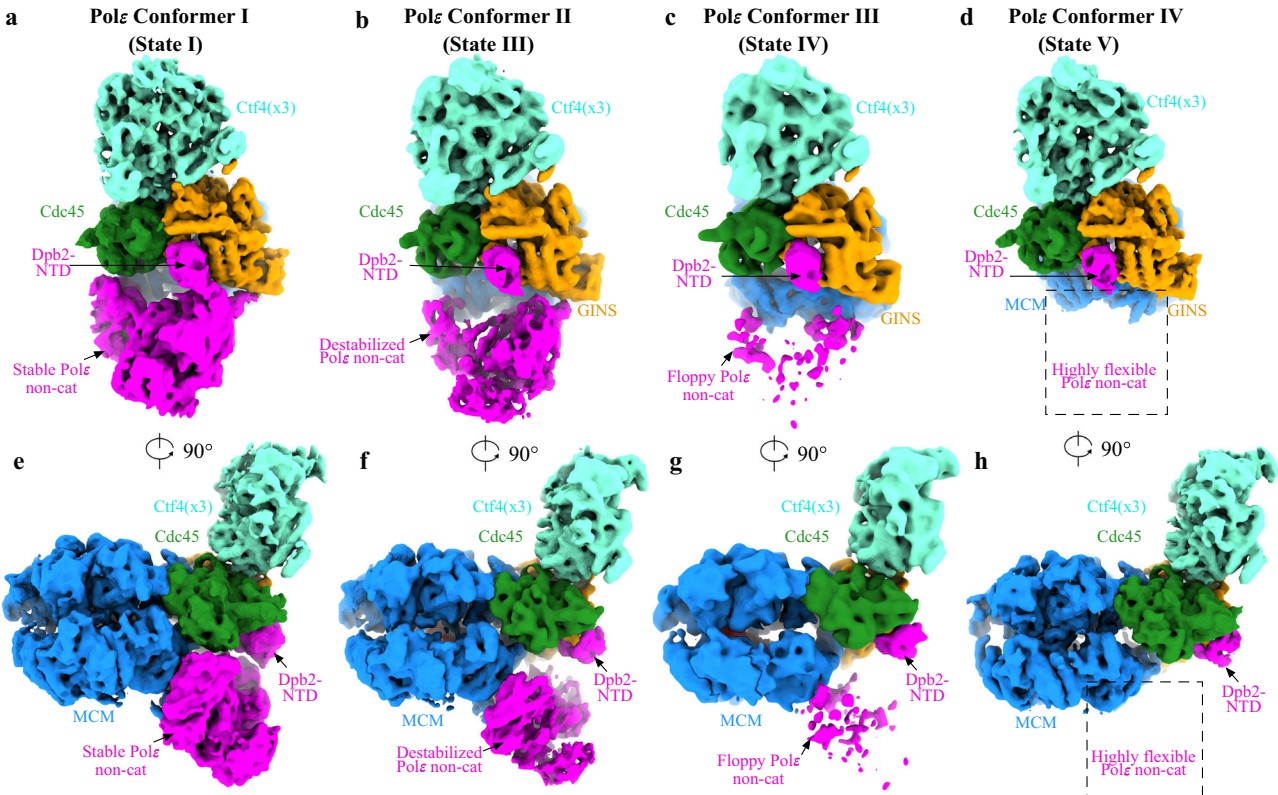

**Fig. 3 | Polε stability in the replisome. a–d** Four different conformations of Polε in the replisome. **e–h** Same as (**a–d**) but with 90° rotation. Replisome components are colored as indicated.

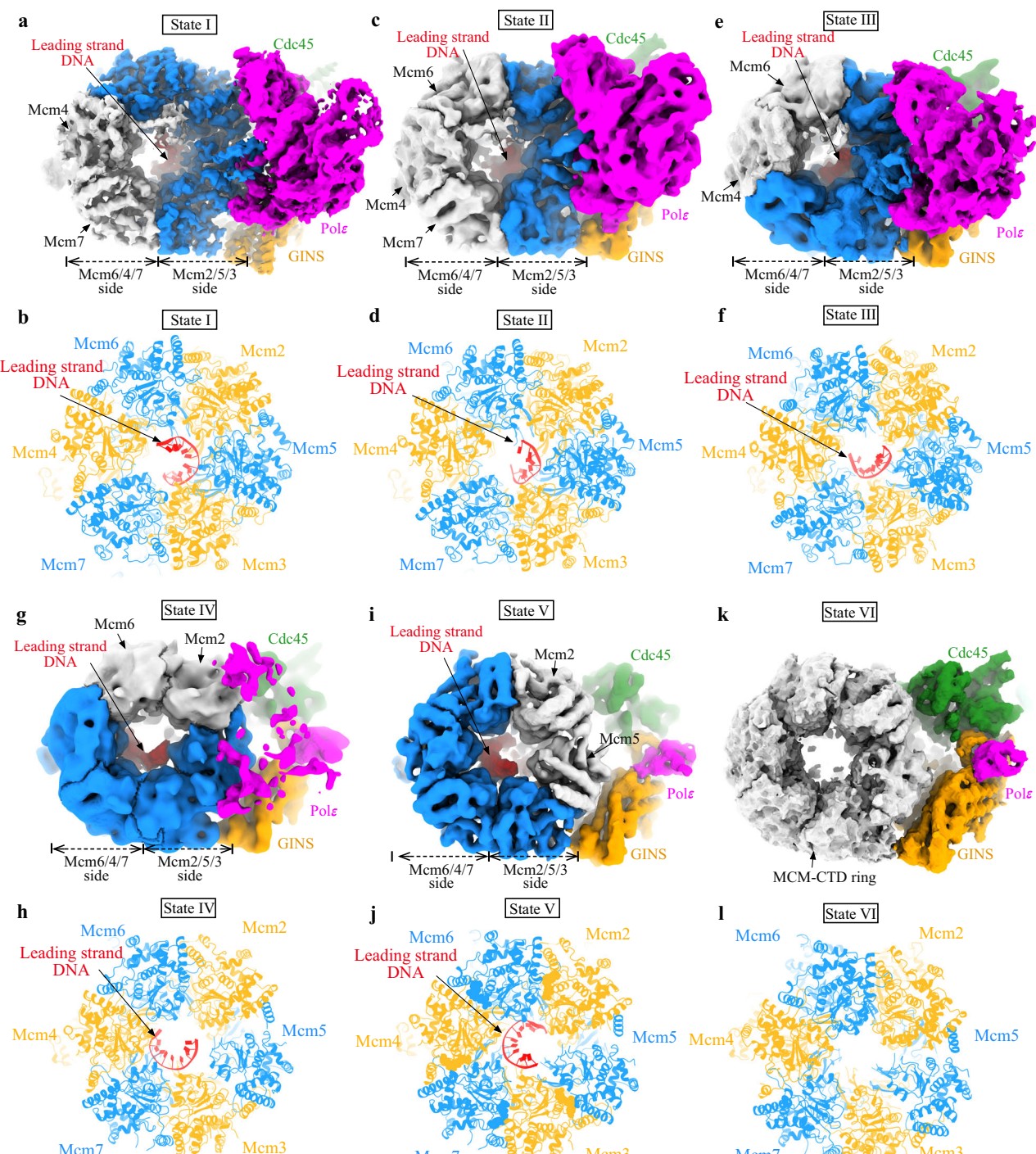

**Fig. 4 | Polε stability and rotational movement of DNA around MCM pore.** Comparison of the leading-strand DNA conformations around MCM pore from different states of the replisome: State I (**a**, **b**), State II (**c**, **d**), State III (**e**, **f**), State IV (**g**, **h**), State V (**i**, **j**), and State VI (**k**, **l**). Relevant density maps (**a**, **c**, **e**, **g**, **i**, and **k**) shown with MCM subunits free of DNA engagement colored in gray. **b**, **d**, **f**, **h**, **j**, and **l** Bottom CTD views of the leading-strand replisome with DNA and Polε in different conformational states (I-V) shown with the atomic models. State VI with no DNA bound in MCM pore.

In State I (3.07 Å), which accounts for 58.5% of the total particles (sum of particles from States I-VI), Polε is stably associated with the MCM ring (Fig. 3a, e), and the leading strand DNA engages with the ATPase loops, pre-sensor1 (PS1) and helix-2-insert (H2I), from Mcm6/2/5/3 subunits arranged in a staircase (Fig. 4a, b, Supplementary Fig. 5). State II (4.23 Å), which accounts for 22.5% of the total particles, is almost identical to State I (Supplementary Fig. 6a), except that the leading-strand DNA is released from Mcm6 (Fig. 4c, d). In State III (2.6% of the total particles, 4.52 Å), the leading-strand DNA is engaged with

Mcm7 while also retaining a weak interaction with Mcm2 (Fig. 4e, f). This state is equivalent to the Conformer II replisome, where Polε appears slightly destabilized (Fig. 3b, f). In State IV (1.2% of the total particles, 7.29 Å), the leading strand DNA is completely released from Mcm2 and translocated onto Mcm4 to engage with Mcm5/3/7/4 subunits (Fig. 4g, h). Coincidentally, Polε is further destabilized in this state (Fig. 3c, g). In State V (4.5% of the total particles, 4.12 Å, equivalent to Conformer IV), the leading-strand DNA becomes disengaged with Mcm5 but is further translocated onto a surface formed by Mcm3/7/4/

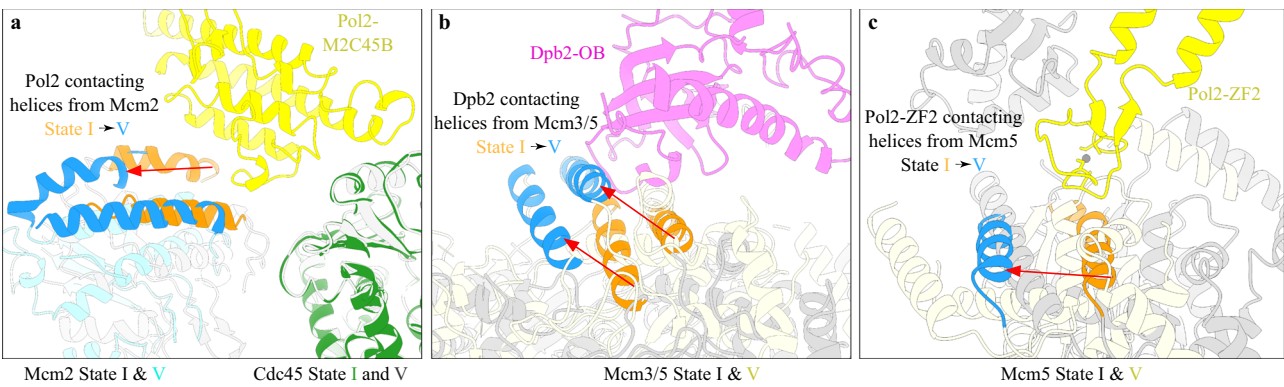

**Fig. 5 | The conformational change in the MCM motor domains regulates Polε docking site formation.** Structural comparison of replisome in State I and State V, showing large displacement in relevant Pol2 (**a**), Dpb2-OB (**b**), and Pol2-ZF2 (**c**) contacting α helices from Mcm2 (**a**), Mcm3:5 interface (**b**) and Mcm5-CTD (**c**).

6 subunits (Fig. 4i, j). At this point, Polε becomes highly dynamic and invisible in the cryo-EM density map (Fig. 3d, h). State VI replisome (10.7% of the total particles, 4.07 Å) shows no obvious electron density corresponding to leading-strand DNA in its MCM pore (Fig. 4k, l), although its overall structure is very similar to that of State V. Our results indicate a strong correlation between DNA translocation and Polε stability (Supplementary Movie 1), suggesting that the interaction between Polε and MCM is important for the coordinated activities of these two engines in the replisome.

To better understand the mechanism regulating Polε stability in the replisome, we compared the CMG structures from different Polε conformers (Supplementary Fig. 7, Supplementary Movie 1). In Conformer I (equivalent to States I and II) with a stable Polε, the MCM ring adopts a compact inter-CTD interface of Mcm2:5 (Supplementary Fig. 7b). However, in Conformer IV (equivalent to State V) where Polε is highly flexible, the same CTD interface appears largely disengaged, despite the domain-swapped helix of Mcm5 still packing with the CTD of Mcm2 (Supplementary Fig. 7h). In agreement with this observation, the measured BSA for the CTD interface of Mcm2:5 is significantly reduced from 6792 Å² in Conformer I to 2381 Å² in Conformer IV (Supplementary Fig. 7b, h). Similar changes can also be observed at the CTD interfaces of Mcm2:6 (Supplementary Fig. 7c, i) and Mcm3:5 (Supplementary Fig. 7a, g). Correspondingly, Polε contacting motifs on Mcm2/5/3 subunits are remodeled during the transition from Conformer I to Conformer IV, largely disrupting the binding surfaces of Polε on the motor domains of the MCM ring (Fig. 5, Supplementary Fig. 7m, n). Therefore, in Conformer IV, Dpb2-NTD becomes the only part connecting Polε to CMG (Fig. 3d, h). Notably, minor conformational changes in the Polε binding sites can also be observed from Conformer II (State III) to Conformer III (State IV) (Supplementary Fig. 6a–d), gradually leading to a floppy Polε (Supplementary Movie 1).

Taken together, these results reveal that Polε's morphology in the replisome changes with the DNA binding mode around the MCM pore. In particular, a stable attachment of the leading-strand DNA to the CTD interface of Mcm2:5 is crucial for modulating the MCM ring into a conformation competent for Polε to dock (Supplementary Fig. 7m, n). These dynamic structures demonstrate how the replisome coordinates DNA translocation by the CMG with Polε binding to the MCM ring in a periodic manner.

## ATPase states in regulating Polε binding to MCM ring
The CMG helicase operates through the C tier of its MCM ring, which serves as a motor powered by ATP binding and hydrolysis in neighboring ATPase active centers to drive DNA threading through its MCM pore[1]. To understand how the activities of those ATPase active centers are coupled to DNA translocation by CMG and Polε activity, we compared the configurations of each ATPase center between State I and

State V of the replisome. In State I, nucleotide binding can be found in five ATPase centers, except for Mcm4:7 (Supplementary Fig. 8). It is noteworthy that three of them (Mcm6:2, Mcm2:5 and Mcm5:3) formed by DNA-engaging subunits are almost identical in a compact configuration with nucleotide bound (Supplementary Fig. 9a, b, f). The electron densities corresponding to the bound nucleotides indicate that Mcm2:5 is bound to ATPγS while Mcm6:2 and Mcm5:3 are bound to ADP (Supplementary Fig. 8a–c). The remaining three ATPase centers (Mcm7:3, Mcm4:7 and Mcm6:4) are less compact with the elements for ATP hydrolysis (sensor 2, sensor 3 and arginine finger) shifted away from the nucleotide-binding motifs (walker A and walker B) in comparison to Mcm2:5 (Supplementary Fig. 9c–e, g). Furthermore, Mcm7:3 and Mcm6:4 are bound to ATPγS (Supplementary Fig. 8d, f), but Mcm4:7 does not show any nucleotide binding (Supplementary Fig. 8e). In State V, DNA spools onto the inner CTD surface of Mcm3/7/4/6 subunits (Fig. 4i, j). At this stage, the ATPase centers of Mcm7:3, Mcm4:7 and Mcm6:4, which are intimately associated with DNA, have a compacted configuration, while the other three become relaxed with Mcm2:5 showing the largest displacement (Supplementary Fig. 9h–l). Although the limited resolution of the density map did not allow the unambiguous identification of the nucleotide bound at each ATPase site, this permutated arrangement of DNA-associated MCM subunits in State V is the opposite of that in State I (Supplementary Figs. 7 and 9). Notably, the ATPase centers of Mcm2:5 from State I and Mcm4:7 from State V are almost identical (Supplementary Fig. 9u), suggesting that these two ATPase sites play crucial roles in driving DNA translocation around the MCM pore. Consistent with this notion, these two active centers are also essential for helicase activation to initiate DNA replication[54].

In addition, there are significant conformational changes in both H2I and PS1 loops between State I and State V (Supplementary Fig. 5). Using Mcm3-CTD as a reference for superimposing the CMGs from these two states, we observe an obvious tilt (-18°) in the State V of the MCM-CTD ring relative to that of State I (Supplementary Fig. 5a, b). These structural differences suggest that orientation changes between adjacent CTDs, induced by nucleotide rearrangement around the MCM ring, can be translated into axial displacement of ATPase loops. This displacement can engage or disengage ssDNA (Supplementary Fig. 5c–h), driving the rotational movement of ssDNA around the MCM pore in vertical steps.

## Leading-strand DNA synthesis by Polε is modulated by its binding to the MCM ring
Physical coupling between helicase and polymerase from phage has been shown to enhance the rates of both DNA unwinding and DNA synthesis[4,6,55]. To determine if Polε-MCM coupling could stimulate each other's activities at the replication fork, we reconstituted both DNA

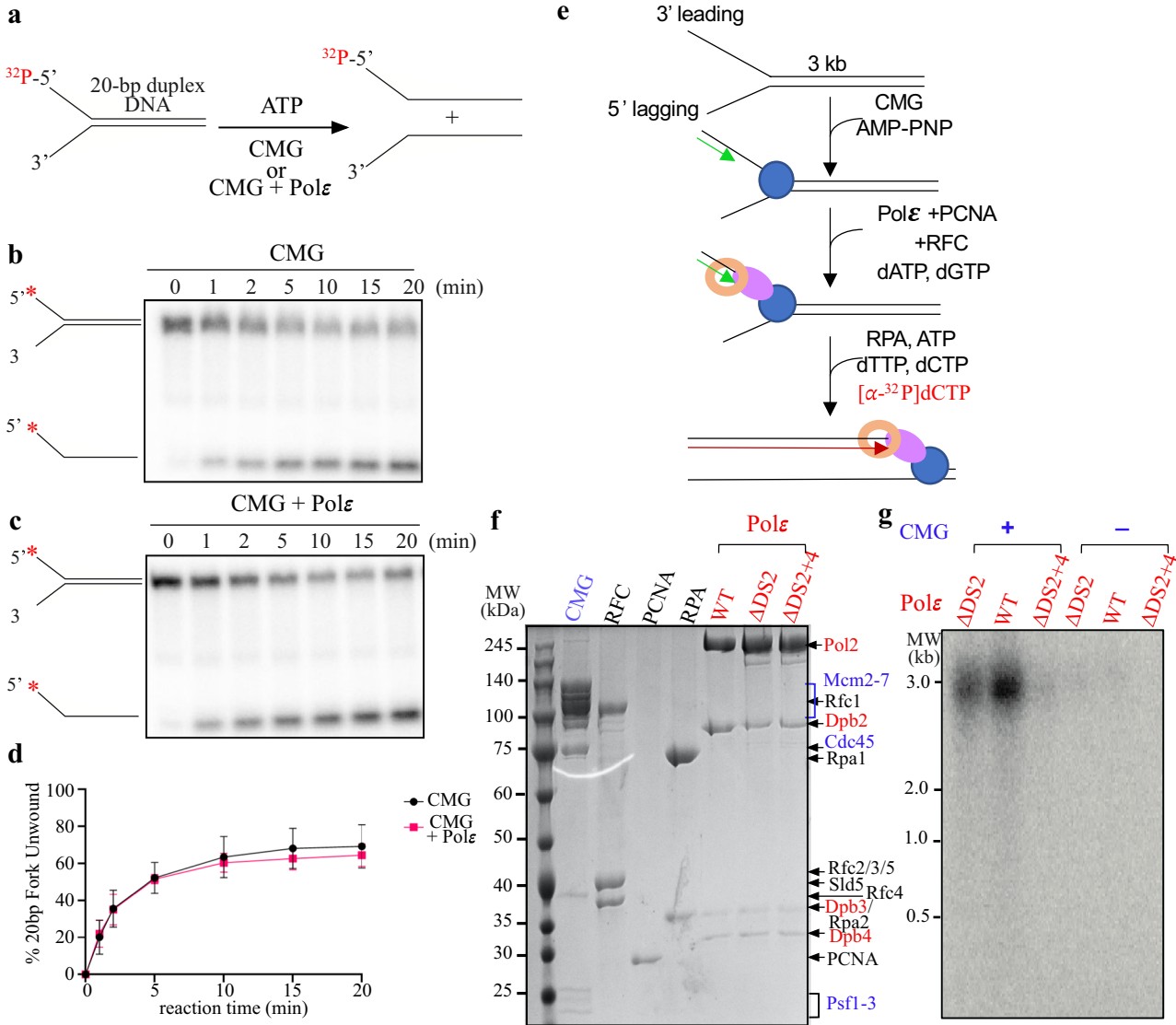

**Fig. 6 | The effect of Polε docking to MCM ring on CMG helicase and leading strand DNA synthesis in vitro. a** Schematic illustration of the unwinding of a 20-bp forked DNA by CMG. **b**, **c** Native PAGE of CMG helicase assays performed as in (**a**) in the absence (**b**) or presence (**c**) of Polε. **d** Quantification of the results; assays were performed in triplicate. Data are mean ± SEM (n = 3). Source data are provided as a source data file. **e** Scheme of reconstituting leading-strand DNA synthesis by Polε (see details in "Methods"). **f** Purified factors used for DNA replication assays analyzed by SDS-PAGE with Coomassie staining. **g** Alkaline agarose gel of reaction products from the replication assays performed as in (**e**). Similar results were obtained in two independent experiments.

unwinding and leading-strand replication with purified yeast proteins in vitro (Fig. 6). First, we examined the effect of Polε on regulating CMG activity using an in vitro helicase assay. In this assay, CMG helicase acted on a synthetic forked DNA with or without Polε (Fig. 6a). Our results showed that in the absence of Polε, there was an efficient unwinding of forked duplex DNA (20 or 60 bp) by CMG helicase (Fig. 6b, d, Supplementary Fig. 10), consistent with a previous study[56]. However, the reactions containing Polε only exhibited a slight suppression of CMG activity (Fig. 6c, d, Supplementary Fig. 10), suggesting that Polε binding has little effect on helicase activity, at least in our in vitro setting.

To examine the impact of Polε-MCM coupling on leading-strand synthesis, we conducted an in vitro replication assay using purified Polε together with CMG, RFC, PCNA and RPA (Fig. 6e, f). First, CMG helicase was pre-loaded onto a DNA substrate which contains a 3.0-kb linear duplex and a synthetic forked junction at one end with its leading strand annealed with a DNA primer in the presence of AMP-PNP. Next, PCNA and RFC, along with either WT or mutant Polε, were

assembled on the DNA substrate with the presence of dATP and dGTP. Replication was initiated by adding the remaining dNTPs, along with [32P]dCTP, RPA and 5 mM ATP to stimulate CMG helicase. In the reaction with the WT Polε, a high level of full-length products was observed (Fig. 6g). However, the control reaction without CMG only showed a very low level of the full-length products, if any. Similar replication was also observed in the reactions with PolεΔDS2 but at a lower efficiency. In contrast, PolεΔDS2+4 did not produce obvious DNA synthesis (Fig. 6g), indicating that disrupting the Polε-MCM coupling suppresses leading-strand DNA replication mediated by Polε in the replisome. Together, our analyses suggest that the binding of Polε to the CTD motor domain of CMG helicase helps to establish leading-strand replication at the replication fork.

## Discussion
Coordinating the activities of replicative helicase and DNA polymerase(s) at the replication fork is crucial for efficient and accurate DNA replication. However, the mechanism underlying this process is

not yet fully understood. In this study, we elucidate the intrinsic mechanism by which DNA translocation by CMG is synchronized with the activities of Polε in the leading strand replisome. By virtue of capturing multiple high-resolution snapshots of a translocating replisome from yeast, we are able to fill in the details of two seemingly contrasting models for the translocation of the CMG helicase and melt their differences into a single model consistent with both sets of data. The first model proposes helicase translocation through a rotary mechanism of threading ssDNA around the MCM inner pore[37,43,57], which places an equal burden on each MCM subunit. The second model proposes a nonsymmetric inchworm mechanism, in which sets of MCM subunits move the ssDNA through the MCM pore, with each MCM subunit having a distinct role in the process[39]. Our results suggest that the CMG helicase assumes dynamic conformations to translocate ssDNA in a rotary fashion though one specific arrangement of the ssDNA bound states (States I) appears to predominate. During DNA translocation, the motor domains of the CMG helicase remain as a planar ring to coordinate DNA threading rather than as an open spiral, as observed in the T7 and *E. coli* helicases[4,58]. In addition, instead of moving its entire motor domain, the CMG helicase generates relatively small shifts in its DNA binding loops (H2I and PS1) to engage or disengage ssDNA by modulating the spacing of its ATPase interfaces (Fig. 4, Supplementary Fig. 5). As some of these states are also observed in *Drosophila* CMG[37], we believe that the CMG helicases use a highly conserved mechanism for DNA translocation in eukaryotes. A unique feature of the nonsymmetric model is that not all ATPase sites contribute equally to DNA translocation by CMG[37]. Our results are consistent with an en bloc shift of two or three ATPase domains during each cycle of ATP binding and hydrolysis to reset their DNA binding loops in the rotary staircase for ssDNA engagement.

The most striking feature of the translocating replisome is the dynamic coupling of Polε with the motion of the CMG helicase at the replication fork. Our findings indicate that the interaction between Polε and the motor domain of CMG helicase is essential for promoting CMG formation during origin firing. However, how Polε drives the assembly of Cdc45 and GINS onto Mcm2-7 double hexamer (DH) remains unclear. As Polε can engage with the two gate-forming subunits, Mcm2 and Mcm5, it is likely that Polε binding reconfigures and/or stabilizes the MCM ring to create a conformation suitable for a stable association of Cdc45 and GINS with the MCM DH. Notably, a recent study reported that human Polε is not required for CMG assembly but is essential at later steps of replication initiation[59], suggesting that the detailed role of Polε in regulating origin firing may be species-specific. Interestingly, once DNA replication is initiated, Polε becomes dispensable for S phase progression in vivo. This is likely due to the ability of Polδ to replace the polymerase function of Polε[9]. However, our in vitro replication assay shows that the Polε-MCM coupling plays a crucial role in maintaining leading-strand DNA synthesis by Polε. Previous studies have shown that Polε exhibits a very low affinity for PCNA[52,60], implying that Polε cycles on and off at ds-ss DNA junctions during DNA synthesis. Under this scenario, it is almost impossible for a replisome dragging a highly flexible Polε to afford a non-stop processive replication on the leading-strand template. We envision that Polε periodically binds to the motor domains of the MCM ring, facilitating efficient re-engagement of the polymerase with nearby PCNA-DNA complex for chain elongation (Fig. 7).

While Polε binding shows little inhibitory effect on CMG function in our in vitro helicase assays (Fig. 6b, c, Supplementary Fig. 10), previous studies have shown that the physical coupling between Polε and CMG can affect the rate of fork progression under certain circumstances[61,62]. For example, the CMG helicase alone can overcome tight protein-DNA roadblocks to travel along fork DNA[62]. However, when Polε is coupled to CMG, the helicase can experience stalling at fork barriers[62]. It has been found that the non-catalytic Pol2-CTD is

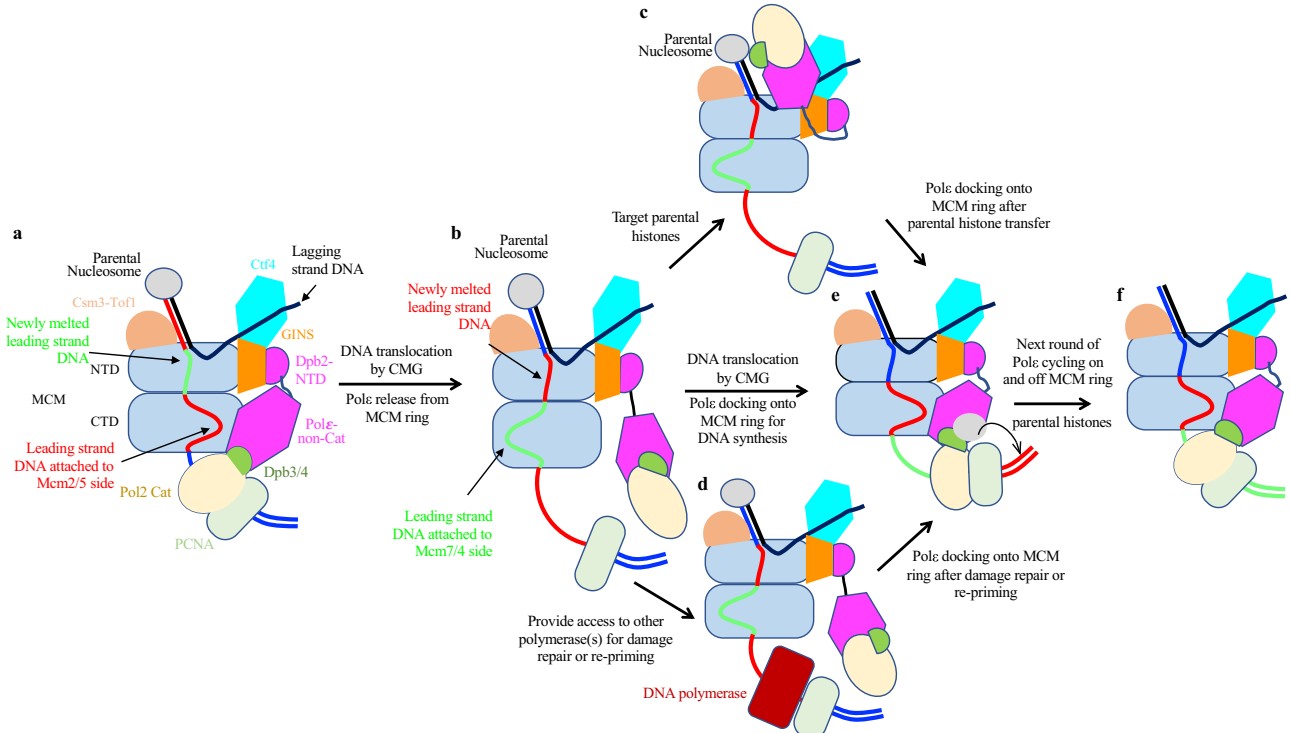

**Fig. 7 | Model illustrating potential functions of Polε cycling on and off MCM ring. a** Polε binding to the MCM ring facilitates its engagement with the PCNA-DNA complex to drive strand extension. **b** DNA translocating away the Mcm2:5 interface releases Polε from MCM. **c** Polε flips over CMG helicase to target parental histones via Dpb3-4 for redisposition of epigenetic information from parental DNA strand to newly synthesized leading strand. **d** Upon DNA damage, a flexible Polε can provide access to other polymerases for damage repair and/or re-priming for fork restart. **e** Leading-strand DNA relocating to the Mcm2:5 interface enables Polε binding to MCM to re-establish strand extension. **f** DNA translocation by CMG initiates a new round of Polε cycling on and off the MCM ring.

responsible for this stalling function of Polε. When encountering fork barriers, it is likely that Pol2 binding to the MCM ring stabilizes the replisome in State I conformation, allowing the helicase to stall until roadblocks are removed by designated helicases such as Pif1 and/or Rrm3[63].

The dynamic behavior of Polε, which undergoes alternating association and dissociation with the MCM ring in the replisome, may be critical for recycling parental histone from the fork front to newly synthesized leading strands. Once released from the MCM ring, Polε can utilize Psf1 as an anchor to leapfrog over the helicase from its C tier to N tier and engage with parental nucleosome or histones (Fig. 7b, c). It is known that Dpb3 and Dpb4 can bind the H3-H4 tetramer directly[64,65]. We envision that the re-engagement of Polε with the MCM ring will facilitate an efficient and accurate transfer of the parental histones captured by Polε to the newly synthesized leading strand (Fig. 7e). However, if this process is interrupted through constantly tethering Polε to the MCM ring during CMG translocation, it could disrupt the normal recycling of parental histones. Further investigation is needed to understand how the Polε-MCM coupling is coordinated with its role in epigenetic inheritance.

In response to DNA damage, the activation of the S phase checkpoint for fork stalling is crucial. The dissociation of Polε from the MCM ring may serve as a potential mechanism to uncouple Polε from CMG. This allows the helicase to continue moving forward, resulting in an excess of ssDNA that triggers S phase checkpoint activation. To maintain fork stability, Rad53 kinase inhibits CMG activity, bringing the replisome to a pause[66]. It has been shown that a damaged DNA template prevents Polε from engaging with the PCNA-DNA complex, which is necessary for DNA synthesis[67]. A flexible Polε that is disengaged from both the MCM ring and DNA template could provide access for other polymerase(s) or factors to repair damage and/or restart the fork (Fig. 7b, d).

In summary, this study presents a dynamic picture of how the CMG helicase and DNA polymerase work together in the replisome to coordinate DNA unwinding with DNA synthesis. This synergy also enables the replisome to overcome a variety of roadblocks and DNA lesions while migrating along the fork to replicate both genetic and epigenetic information. As similar replisome architectures are found in bacteriophage T7, *E. coli* and yeast, the periodic coupling between helicase motor domains and DNA polymerase may serve as a universal strategy that living organisms use for high-fidelity genome duplication.

## Methods

### Purification of CMG
CMG was purified with a yeast strain, yJCZ2, in which FLAG tags were fused to the N-terminal regions of ectopically expressed Mcm3 and Cdc45, respectively, as previously described[68] with modifications. Cells were cultivated in YPA supplemented with 2% raffinose until $OD_{600}$ reached 2.0 and then induced for protein expression with 2% galactose for 3 h at 30 °C. Cells were harvested and washed with buffer A (100 mM KCl, 25 mM HEPES pH 7.6, 10% glycerol, 4 mM MgCl₂, 2 mM β-ME, 0.02% NP-40) and suspended with 50 ml of buffer A containing 2×protease inhibitor (PI) cocktail (Roche, 05056489001). The cell suspension was frozen dropwise in liquid nitrogen and then disrupted with a freezer mill (SPEX CertiPrep 6850 Freezer/Mill). The cell powder was thawed on ice by adding an equal volume of buffer A with 1×PI and 1 mM phenylmethylsulfonyl fluoride (PMSF; Thermo, 36978). The cell lysate was clarified by centrifugation at 32,300×g for 20 min at 4 °C. Then, anti-FLAG M2 affinity resin (Sigma, A2220) was incubated with the supernatant for 3 h at 4 °C. The beads were then washed with 30 column volume (CV) of buffer A. Proteins were eluted with 1 mg/ml of 3×FLAG peptide (GenScript, U6320GJ210-1) in buffer A. The eluate was then loaded onto a Mono Q 5/50 GL (Cytiva, 10223365), equilibrated with buffer A and eluted with a continuous gradient of KCl from 100 to 550 mM. The fractions containing the intact CMG were combined and

concentrated before being subjected to a Superose 6 increase 10/300 GL column (Cytiva, 29091596) equilibrated in buffer A. Peak fractions containing the intact CMG were pooled, aliquoted, and stored at −80 °C.

### Purification of the wild-type and mutant DNA polymerase ε
Wild-type DNA polymerase ε was purified from yAJ2 containing a CBP tag at the N-terminus of ectopically expressed Pol2 as previously described[19] with modifications. Cells were grown in YPA supplemented with 2% raffinose at 30 °C and induced for protein expression with 2% galactose for 3 h when $OD_{600}$ reached 2.5. Cells were harvested and washed twice with buffer B (400 mM potassium acetate, 25 mM HEPES pH 7.6, 10% glycerol, 2 mM β-ME, 0.01% NP-40). The cells were lysed by a freezer mill and thawed by adding an equal volume of buffer B supplemented with 1×PI and 1 mM PMSF. The supernatant was clarified by centrifugation at 32,300×g at 4 °C for 20 min and then incubated with calmodulin affinity resin (Agilent, 214303) in the presence of 2 mM CaCl₂ at 4 °C for 3 h. After extensive washing, polymerase ε was eluted with TEV protease treatment (100 µg/ml of TEV) for 6 h at 4 °C. The collected sample was then applied to a Mono Q 5/50 GL equilibrated with buffer B and eluted with a continuous gradient of potassium acetate gradient from 400 to 1000 mM. Peak fractions were pooled and concentrated before being subjected to a Superose 6 increase 10/300 GL column equilibrated in buffer B. Peak fractions were pooled, aliquoted and stored at −80 °C.

The mutant Polε assemblies containing Pol2ΔDS2 or Pol2ΔDS2+4 along with Dpb2-WT or Dpb2ΔNTD were purified from strains yXZ1-6 (see Supplementary Table 2 for details) in the same manner as the WT Polε.

### Purification of Tof1-Csm3
Tof1-Csm3 was purified from strain yAE48 containing a CBP tag at the N-terminus of ectopically expressed Csm3 as previously described[8,37] with modifications. Cells were cultured in a YPA medium supplemented with 2% raffinose and induced for protein expression with 2% galactose when $OD_{600}$ reached 2.5. The cells were collected and washed twice with buffer C (200 mM NaCl, 25 mM HEPES pH 7.6, 10% glycerol, 2 mM β-ME, 0.02% NP-40). The cell pellet was then suspended with 50 ml of buffer C supplemented with 2×PI and frozen dropwise in liquid nitrogen for further disruption by the freezer mill. The cell powder was thawed on ice by adding an equal volume of buffer C supplemented with 1 mM PMSF and 1×PI. After centrifugation clarification at 32,300×g at 4 °C for 20 min, calmodulin affinity resin was added into the supernatant for 3-h incubation at 4 °C in the presence of 2 mM CaCl₂. The resultant resin was extensively washed with buffer C containing 2 mM CaCl₂. Tof1-Csm3 was eluted with TEV protease treatment for 6 h at 4 °C. The collected sample was applied to a Mono Q 5/50 GL equilibrated with buffer C and eluted with a continuous gradient of NaCl from 200 to 400 mM. Peak fractions were pooled and concentrated for further purification with a Superose 6 increase 10/300 GL column equilibrated in buffer C. Peak fractions were pooled, aliquoted and stored at −80 °C.

### Purification of Ctf4
Ctf4 was purified from an *E. coli* BL21(DE3) Rosetta strain (pXZ1) bearing pET28-6xHis-Sumo-CTF4 as previously described[69] with modifications. Overexpression of Ctf4 was induced by 1 mM of isopropyl β-D-thiogalactoside (IPTG; ChemCruz, SC-202185C) at 18 °C for about 16 h when cell density reached an $OD_{600}$ of 0.6. The cell pellet was suspended in buffer D (300 mM NaCl, 50 mM Tris pH 8.0, 10% glycerol, 0.01% NP-40, 2 mM β-ME) supplemented with 1×PI and 1 mM PMSF. Cell lysates were prepared with a high-pressure homogenizer (Union, UH-06). The clarified supernatant was then loaded onto Ni-NTA affinity beads (Biotech, F023KA2021) and washed for 50 CV of buffer D containing 40 mM of imidazole. The proteins were eluted with

3 CV of buffer D containing 250 mM of imidazole. The eluates were then dialyzed in buffer E (150 mM NaCl, 25 mM HEPES pH 7.6, 10% glycerol, 0.01% NP-40, 2 mM β-ME) before being treated with sumo protease for ~18 h at 4 °C. Afterward, the protein sample was reloaded onto Ni-NTA affinity beads to remove affinity tags. The flow through containing Ctf4 protein was then applied to a Mono Q 5/50 GL equilibrated with buffer E and eluted with a continuous gradient of NaCl from 320 to 380 mM. Peak fractions were processed and further purified with a Superose 6 increase 10/300 GL column equilibrated in buffer E. Ctf4 fractions were pooled, aliquoted and stored at −80 °C.

### Purification of RPA
RPA was purified from a yeast strain yAE31 containing a CBP tag at the N-terminus of Rfa1 as previously described[19] with modifications. Cells were grown in YPA supplemented with 2% raffinose at 30 °C and induced for protein expression with the addition of 2% galactose when $OD_{600}$ reached 3.0. The cells were collected and washed twice with buffer F (300 mM NaCl, 25 mM HEPES pH 7.6, 10% glycerol, 0.02% NP-40, 2 mM β-ME). The cell pellet was resuspended with 50 ml of buffer F supplemented with 1×PI and then frozen dropwise in liquid nitrogen for further cell disruption by freezer mill. The cell powder was thawed on ice by adding 50 ml of buffer F supplemented with 1×PI and 1 mM PMSF (final concentration). After centrifugation clarification to remove insoluble debris at 32,300×$g$ for 20 min at 4 °C, 2 ml of cal-modulin affinity resin was added into the recovered supernatant for incubation at 4 °C for 3 h in the presence of 2 mM CaCl₂, followed by extensive washing of the resin with buffer F containing 2 mM CaCl₂. RPA was then eluted with buffer F supplemented with 2 mM EDTA and 2 mM EGTA. The eluate was further applied to a Mono Q 5/50 GL. Peak fractions were pooled and concentrated by further purification with a Superose 6 increase 10/300 GL column equilibrated in buffer F. Peak fractions containing RPA were pooled, aliquoted, and stored at −80 °C.

### Purification of RFC
RFC was co-expressed and purified from an *E. coli* BL21(DE3) Rosetta strain (pXZ2) harboring RFC1-5 subunits as previously described[70] with modifications. Overexpression of RFC subunits was induced by 0.5 mM of ITPG for 16 h at 15 °C when cell density reached an $OD_{600}$ of 0.5. The cell pellet was suspended in buffer G (150 mM NaCl, 25 mM HEPES pH 7.6, 10% glycerol, 2 mM β-ME) supplemented with 1×PI and 1 mM PMSF. The cell lysate was prepared with a high-pressure homogenizer. The supernatant was clarified by centrifugation at 32,300×$g$ at 4 °C for 1 h and then loaded onto a 1-ml HiTrap SP FF column (GE Healthcare, 17-5054-01) equilibrated with buffer G. Elution was performed with a continuous gradient of NaCl from 150 to 600 mM. The peak fractions containing RFC were pooled and dialyzed in buffer H (300 mM NaCl, 25 mM HEPES pH 7.6, 10% glycerol, 2 mM β-ME). Afterward, the protein sample was further processed with a Mono Q 5/50 GL equilibrated with buffer H. The flow-through fractions were pooled, concentrated, and applied to a Superdex 200 column equilibrated in Buffer H. Final peak fractions were pooled, aliquoted, and stored at −80 °C.

### Purification of PCNA
PCNA was purified from an *E. coli* BL21(DE3) Rosetta strain (pXZ3) containing a pET28a-6xHis-PCNA. PCNA overexpression was induced with 0.5 mM IPTG for 18 h at 18 °C when cell density reached an $OD_{600}$ of 0.6. The cell pellet was resuspended in buffer I (150 mM NaCl, 25 mM HEPES pH 7.6, 10% glycerol, 2 mM β-ME) supplemented with 1×PI and 1 mM PMSF. The cell lysate was prepared with a high-pressure homogenizer. The clarified supernatant was passed through a 2-ml bed volume of Ni-NTA resin, followed by extensive washing with buffer I containing 40 mM of imidazole. Elution was conducted with buffer I containing 250 mM of imidazole. The protein sample was further applied onto a Mono Q 5/50 GL equilibrated with buffer I and eluted with a continuous gradient of NaCl from 150 to 600 mM. Peak fractions

were concentrated and loaded onto a Superdex 200 column equilibrated in Buffer I. Final peak fractions were pooled, aliquoted, and stored at −80 °C.

### Preparation of fork DNA for complex reconstitution
Leading strand sequence is 5′(Cy3)TAGAGTAGGAAGTGATGGTAAGTG ATTAGAGAA TTGGAGAGTGTG(T)₃₄ T*T*T*T*T*T, where * represents phosphorothioate backbone linkages. Lagging strand sequence is 5′GGCAGGCAGGCAGGCACACACTCTCCAAT TCTCTAATCACTTAC-CATCACTTCCTACTCTA. Both oligos were dissolved in 10 mM Tris pH 8.0 to a final concentration of 100 μM respectively and then mixed with a 1:1 ratio for DNA annealing with a thermocycler. DNA was denatured at 95 °C for 5 min, and then the temperature was decreased to 4 °C at the rate of 0.1 °C/s.

### Replisome reconstitution
Replisome was reconstituted in vitro as previously described[41] with some modifications. Then, 200 μl of 0.7 μM CMG was mixed sequentially at room temperature with 1.5 molar excess of other components according to the order shown in Supplementary Fig. 1a. The assembly mixture was adjusted to contain 100 mM KCl, 25 mM HEPES pH 7.6, 4 mM MgCl₂ and 4 mM ATPγS (Roche, 11162306001). The mixture was then subjected to a 2 ml of 10−30% glycerol gradient containing glu-taraldehyde (Sigma, G5882) (0−0.16%) for fixation in a buffer (100 mM KCl, 25 mM HEPES pH 7.6, 4 mM MgCl₂, 0.5 mM ATPγS), and separated by ultracentrifugation (Beckman Optima TLX TLS55 rotor) at 200,000×$g$ for 2 h at 4 °C. The fractions containing intact replisomes according to silver staining results with the samples without cross-linking were pooled and processed for EM analyses.

### In vitro binding assay
CBP-tagged WT and mutant Polε (52 pmol) were pre-incubated with a 10-μl bed volume of calmodulin affinity resin in buffer B supplemented with 1×PI for 4 h at 4 °C. The resultant resin was washed with a low salt buffer (150 mM KAc, 25 mM HEPES pH 7.6, 4 mM MgCl₂, 2 mM CaCl₂, 0.02% NP-40, 10% glycerol, 1xPI) in the presence of 0.5 mM of ATPγS. CMG alone or CMG (19.1 pmol) pre-mixed with a 16-dT (TTTTTTTTTTTTT*T*T*T) ssDNA substrate (41.67 pmol) were then incubated with the resin for 3 h at 4 °C. (* represents phosphorothioate backbone linkage modification to prevent digestion by Polε's exonu-clease activity). Afterward, the resin was precipitated and washed extensively with the abovementioned low salt buffer. The resin was resuspended in SDS loading buffer and heated to 95 °C for 5 min. Proteins were separated by 10% Bis-Tris polyacrylamide gel and visualized by Coomassie staining.

### Yeast cell growth assay
Yeast strains used for cell growth assay were isogenic to YST3085 (*pol2-iAID*) (see Supplementary Table 2 for details). pRS403 constructs bearing either WT *POL2* or *pol2* mutants under the native promoter and terminator of *POL2* were integrated into his3 locus of YST3085, respectively. The cells of indicated strains were examined on YPAD agar plate or YPAD agar plate containing 1 mM 1-naphthalene acetic acid (NAA; Sigma, N0640) and 20 μg/ml Doxycycline (Sigma, D9891) at 30 °C.

### Flow cytometry analysis
Standard methods were used to synchronize yeast cells in the G1 phase using α factor (GenScript, U187ege180-1). To deplete Pol2-AID pro-teins, 1 mM NAA and 20 μg/ml doxycycline were added to G1 cells for 2 h at 30 °C. Cells were then washed and released into a fresh medium containing 1 mM NAA, 20 μg/ml doxycycline and 0.05 g/L Pronase (Roche, 33902321). Samples were collected at indicated time points.

To block cells at the early S phase, 200 mM of hydroxyurea (HU; Dalian Meilun Biotechnology, MB1307) was added to cells released

from pre-G1 synchrony and cultured for 1 h. After Pol2-AID depletion, the cells were released into a fresh medium containing 1 mM NAA and 20 μg/ml doxycycline at 30 °C. α factor was then added to arrest cells at the next G1 phase. Samples were collected at indicated time pointsFlow cytometry analyses were performed using a standard procedure. Ethanol-fixed yeast cells were washed with and resuspended in sodium citrate solution. RNase (Thermo Scientific, EN0531) and Proteinase K (Thermo Scientific, EO0491) were added sequentially to remove RNAs and proteins. DNA was stained with PBS buffer (pH 7.4; 137 mM NaCl, 2.7 mM KCl, 10 mM $Na_2HPO_4$, 1.8 mM $KH_2PO_4$) containing 0.5 mg/ml Propidium iodide (Sigma, P4170). Flow cytometer was performed using BD FACSAriaIII Flow Cytometer and analyzed by FlowJo software.

## Immunoprecipitation assay

For this, 500 ml of cell culture for each sample was collected and washed in 25 ml of WCE buffer (100 mM Hepes-KOH pH 7.9, 200 mM potassium acetate, 10 mM magnesium acetate, 2 mM EDTA, 2 mM sodium fluoride) at 4 °C. Cell pellet was then resuspended in one-third volumes of 4× WCE buffer supplemented with 1 mM dithiothreitol (DTT), 1 mM PMSF and 1× PI and frozen dropwise in liquid nitrogen for further cell disruption by freezer mill. The cell power was thawed on ice and diluted to 1x WCE buffer. Then, 1000 U/ml Benzonase Nuclease (Beyotime, D7126) was added to solubilize chromatin-bound proteins for 1.5 h at 4 °C. After centrifugation clarification at 21,500×$g$ for 10 min at 4 °C, the soluble fraction was incubated with anti-FLAG M2 affinity gel for 2 h at 4 °C. Proteins were analyzed by SDS-PAGE and immunoblotting. Antibodies used in probing western blots include anti-Flag (1:1000, Cell Signaling Technology, #14793), anti-Cdc45 (1:5000, a gift from Bruce Stillman), and anti-Mcm2 (1:1000, a gift from Bruce Stillman).

## Sample vitrification

For this, 3 μl of fresh sample was applied to glow discharged holey carbon grids (C-flat R 1.2/1.3 Au) and plunge frozen in liquid ethane cooled by liquid nitrogen using Vitrobot VI (Thermo) after 30 s blotting with filter paper at 4 °C and 100% humidity.

## Electron microscopy data acquisition

The grids were loaded onto an FEI Titan Krios G3i transmission electron microscope operated at 300 kV. Images were recorded on a Gatan K3 summit direct electron detector and a Bio Quantum energy filter with a 20 eV slit width. Images with a total dose of 53 e⁻/Å² were acquired within 4.5 s at a nominal magnification of 81,000 (EFTEM mode), corresponding to a physical pixel size of 1.06 Å. Dose were fractionalized to 40 frames equally. The defocus range was set between −1.0 and −2.5 μm. EPU software v2.10.0 was used for data collection.

## Data processing

MotionCor2 v1.4.0[71] was used for 7×5 patches on motion correction with dose weighting to generate a summed image for each movie. Motion-corrected sums without dose-weighting were used for contrast transfer function (CTF) estimation with Gctf[72]. The image quality and the fitness of CTF were examined manually. Motion-corrected sums with dose-weighting were used for all other image processing. Particles were picked automatically by Gautomatch (https://www.mrc-lmb.cam.ac.uk/kzhang/Gautomatch/) and extracted by RELION 3.1[73]. The selected particles were then imported into cryoSPARC v2.15.0[74] for 2D classifications, ab initio reconstructions, 3D homogeneous and non-uniform refinements and classifications. Well-sorted particles were then recentered, reextracted and combined in RELION. The combined particles (769,392 in total) were imported to cryoSPARC again for further analysis. As shown in Supplementary Fig. 1, heterogeneous refinement generated three main populated classes (Supplementary

Fig. 1g). Particles from the two classes containing strong Polε density were combined for 2D classification to sort out good particles, generating a high-resolution map by non-uniform refinement (Supplementary Fig. 1h). With this map, we performed masked classifications without alignment in RELION focusing on the MCM5-WHD (Supplementary Fig. 1n), MCM-CTD (Supplementary Fig. 1m) and Polε-MCM interface respectively. Meanwhile, this refinement was also performed with Polε-CMG interfaces (Supplementary Fig. 1i) and generated a map with clear densities for the relevant interfaces. By low pass filtering (15 Å), the map showed extra densities corresponding to Tof1-Csm3 and the catalytic domain of Polε at a low-density threshold level in USCF Chimera v1.11.2. To better understand the conformations of these extra densities, we subtracted the densities of Mcm2/3/5, Cdc45, GINS and the non-catalytic domain of Polε for locally refined particles in cryoSPARC. Then skip-alignment classifications were performed with those subtracted particles in RELION, focusing on the above-mentioned extra densities corresponding to Tof1-Csm3 and the catalytic domain Polε (Supplementary Fig. 1j, k). The classification information was mapped back to the original particles following a popular subtraction-based classification protocol[75]. Metafiles describing particle information were converted from cryoSPARC csv version to relion star version by UCSF pyem[76]. A homemade Python script was used to map the classification information for the subtracted particles to the original ones.

To identify different conformations of the motor domain of the MCM ring, similar skip-alignment classifications focusing on the CTD of Mcm2-7 were performed with the particles showing both structured and flexible Polε, respectively. This analysis identified six different states of the MCM-CTD in DNA engagement (Supplementary Fig. 1l, m). A similar classification protocol was also conducted with Mcm5-WHD and generated a series of classes with this domain in various degrees of stability (Supplementary Fig. 1n). All classifications were non-uniformly refined by cryoSPARC to confirm the classifications and inspect the overall structures. The local resolution map was calculated using cryoSPARC.

## Model building and refinement

Ab initio model building was carried out in Coot v0.9.5[77] and PHENIX v1.20[78]. The structure of the CMG bound with Tof1-Csm3 and Ctf4 (PDB: 6skl) was used as an initial model against the map of the State I replisome. The main body of the replisome model was built mainly based on the State I map (EMD-37211, Supplementary Fig. 1m). To facilitate model building and structural analysis for Pol2, Dbp2 and GINS, focused refinement was applied to improve the local resolution of the relevant regions (Supplementary Fig. 1i). These maps were deposited as additional maps of EMD-37211. The models were refined against the corresponding maps using real-space refinement in PHENIX[78]. For other models representing state II-V, the state I model (PDB: 8KG6) was fitted into the corresponding cryo-EM maps. After deleting those subunits missing in the relevant cryo-EM densities, a rigid alignment was performed in Chimera to match with the conformational changes but without adjusting side chains. Due to the limited resolution, the models of the States II-IV replisomes were built with the rigid_body refine (PHENIX) and secondary structure restraints. UCSF Chimera v1.11.2[79], UCSF Chimera X v1.3[80], and PyMOL Molecular Graphics System (version 1.8.X Schrodinger, LLC) (http://pymol.org) were used for figure preparation. The buried surface areas were calculated by PISA (http://www.ebi.ac.uk/pdbe/prot_int/pistart.html).

## Helicase assay

Forked DNA substrate was prepared as previously described[56]. For the 20-bp forked DNA, the lagging ssDNA (GGCAGGCAGGCAGGCAGGCAGGCAGGCAGGCAGGCAGGCAGGTTGGCGAATTCCCA*T*T*G) was labeled by ³²P-γ-ATP using T4 PNK (NEB, M0201S). The forked DNA

was annealed with 1 µM of labeled lagging ssDNA and 5 µM of leading ssDNA(CAATGGGAATTCGCCAACCTTTTTTTTTTTTTTTTTTTTTTTTT TTTTTTTTTTTTT*T*T*T). To perform helicase assay, 350 nM of CMG was first incubated at 30 °C with 5 mM of ATP and 350 nM of Polε in a reaction buffer (100 mM NaCl, 25 mM HEPES 7.6, 10% glycerol, 10 mM MgCl$_2$, 5 mM β-ME and 0.02% NP-40). The forked DNA was then added to a final concentration of 10 nM to start the reaction. Aliquots of the reactions were collected at indicated times and quenched by adding 1% SDS and 20 mM EDTA.

For the 60-bp forked DNA, the lagging ssDNA (1 µM) (GGCAGGCA GGCAGGCAGGCAGG CAGGCAGGCAGGCAGGCAGGTTGGCGAATTCC CATTGCCTCAGCAGATATCACCTCAGCACGACGTTGTAAAAC*G*A*G) was labeled and annealed with the leading ssDNA (5 µM) (CTCGTT TTACAACGTCGTGCTGAGGTGATATCTGCTGAGGCAATGGGAATTCG CCAACCTTTTTTTTTTTTTTTTTTTTTTTTTTTTTTTTTTTTT*T*T*T). The annealed forked DNA was then gel purified from a native 10% PAGE. To perform helicase assay, 350 nM CMG was first incubated at 30 °C with 5 mM ATP and 350 nM Polε in a reaction buffer (100 mM NaCl, 25 mM HEPES (PH 7.6), 10% glycerol, 10 mM MgCl$_2$, 5 mM β-ME and 0.02% NP-40). The 60-bp forked DNA was added to a final concentration of 10 nM to start the reaction. Then, 1 µM unlabeled oligo (AGGTTGGCGAATTCCCATTGCCTCAGCAGATATCACCTCAGCACGAC GTTGTAAAAC*G*A*G) was added immediately after initiating the reaction to prevent reannealing of unwound radiolabeled lagging DNA. Aliquots of the reactions were removed at indicated times and quenched by adding 1% SDS and 20 mM EDTA, followed by flash freezing in liquid nitrogen. All samples were quickly thawed and incubated with Proteinase K at 37 °C for 10 min.

All samples were fractionated on a 10% native PAGE. The gel was then exposed to phosphor screen (Azure Biosystems). The screen was scanned by Sapphire Biomolecular Imager (Azure Biosystems). Quantification and statistical analysis of the figures were performed using ImageJ and GraphPad Prism 8.0 software.

### In vitro replication assay
The fork DNA substrate was prepared by linearizing pUC19 plasmid with BsaI-HF (NEB, R3733L) and then ligated with a preformed fork DNA (leading ssDNA: ACCGAGGTGGTGGAGGCAGGAGTGTTTTTTTTTT TTTTTTTTTTTTTTTTTTTTTTTTTTTTTTGTGTGTGTTGCTCCTGTG GTGGGTGGTGAGAGGAGG, a lagging ssDNA: TTTTTTTTTTTTTTTTT TTTTTTTTTTTTTTTTTTTTTTTTTTTTTTCACTCCTGCCTCCACCA CCT, and primer DNA: TCCTCCTCTCACCACCCACCACAGGAG)[52]. The unligated fork DNA was removed with a Mono Q 5/50 GL with gradient of NaCl from 550 mM to 800 mM. The DNA substrate was then precipitated by 70% ethanol and resuspended with 10 mM of Tris (pH 8.0). Replication assays were performed with 2.5 nM forked DNA substrates, 125 nM CMG, 500 nM RPA, 100 nM PCNA, 100 nM RFC, and 100 nM Polε (WT or mutants) in a reaction buffer (100 mM KOAc, 25 mM HEPES, pH 7.6, 10 mM Mg(OAc)$_2$, 0.1 mM AMP-PNP, 60 µM dATP, 60 µM dCTP, 60 µM dGTP and 60 µM dTTP, 30 uCi of [α−32P]dCTP, 5 mM ATP, 2 mM β-ME, 0.01% NP-40 and 50 µg/ml BSA). First, CMG was pre-incubated with DNA in the presence of 0.1 mM AMP-PNP (Sigma, A2647) for 20 min at 30 °C. Then, PCNA, RFC, Polε (WT or mutants), dGTP, and dATP were added and incubated for 2 min. DNA synthesis was initiated by the addition of RPA, dCTP, dTTP, [α-32P]dCTP, and ATP. After 16 min of incubation at 30 °C, the reactions were quenched by the addition of 0.5% SDS and 40 mM EDTA (final concentration) and analyzed on 1% alkaline agarose gel. Dried gels were exposed to phosphor screen (Azure Biosystems). The screen was scanned by Sapphire Biomolecular Imager (Azure Biosystems).

### Reporting summary
Further information on research design is available in the Nature Portfolio Reporting Summary linked to this article.

## Data availability
Cryo-EM density maps of the replisome State I-V and related supported maps have been deposited in the Electron Microscopy Data Bank (EMDB) with accession codes EMD-37211, EMD-37213, EMD-37215, EMD-37345, and EMD-37343 respectively. Atomic coordinates of the replisomes have been deposited in the Protein Data Bank (PDB) with accession codes 8KG6, 8KG8, 8KG9, 8W7S and 8W7M respectively. Source data are provided with this paper.

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

## Acknowledgements

We thank John Diffley and Karim Labib for yeast strains; Yingyi Zhang at the Biological Cryo-EM Center and Paul Weile Luo at the Information Technology Services Center at the Hong Kong University of Science and Technology (HKUST), Yongqian Zhao (HKUST), Clare S. K. Lee and Elisa Lau (HKU) for technical assistance; Bik Tye and Xiaolan Zhao for comments to the manuscript; and members of our laboratories for discussion. We thank the Biological Cryo-EM Center, generously supported by a donation from the Lo Kwee Seong Foundation at HKUST for data collection. The work was supported by the Research Grants Council (RGC) of Hong Kong (GRF17112119, C7028-19GF, C7009-20GF, GRF17119022, and GRF17109623 to Y.L.Z., ECS26101919, GRF16103321, C7009-20GF and C6001-21EF to S.D., C7028-19GF, to X.L.) and the National Natural Science Foundation of China (32170548 to T.I.). S.D. also acknowledges support from the Hong Kong Branch of Southern Marine Science and Engineering Guangdong Laboratory (Guangzhou) (SMSEGL20SC01-L), Guangdong Basic and Applied Basic Research Foundation (2021A1515012460), Shenzhen Special Fund for Local Science and Technology Development Guided by Central Government (2021Szvup140) and HKUST start-up and initiation grants. D.Y. is supported by an LKS fellowship.

## Author contributions

Y.Z. and S.D. conceived the project. Z.X. purified all proteins and assembled the replisome complex; S.D., J.F., and Z.X. prepared cryo samples; S.D. and J.F. collected cryo-EM datasets; D.Y., J.F. and S.D. processed images; Y.Z., S.D., and J.F. built the atomic models; Y.H., Z.X., and X.M. conducted functional assays; Y.Z., S.D., T.I., D.Y., J.F., Z.X., H.Y., X.M., W.L., Z.L., and X.L. analyzed the data, prepared the figures and wrote the manuscript.

## Competing interests

The authors declare no competing interests.
