## [Peer Review File · Nature Communications]

Synergism between CMG Helicase and Leading Strand DNA Polymerase at Replication ForkEditorial Note: Parts of this Peer Review File have been redacted as indicated to remove third-party material where no permission to publish could be obtained.

REVIEWER COMMENTS

Reviewer #1 (Remarks to the Author):

Summary. Replication forks contain three different molecular motors: the helicase (CMG complex), and leading and lagging strand DNA polymerases (epsilon and delta). Although each of these motors have independent activities, these activities synergistically change when the motors are combined on a single DNA substrate. However, a molecular explanation for this synergy is currently lacking. Using purified proteins and a forked DNA substrate, Xu et. al., reconstituted physiologically relevant leading strand complexes containing CMG and Pol epsilon. Through the inclusion of ATPγS, a limited array of conformational isomers was formed that presumably reflect different snap-shots in the leading strand reaction cycle. Using high-resolution Cryo EM, the resulting structural models demonstrated that pol epsilon is attached to the CMG complex at multiple sites. Interestingly, different attachment patterns between pol epsilon and CMG varied among the structures and correlated strongly with DNA attachment to specific CMG subunits and nucleotide occupancy in specific CMG active sites. These correlations implied that the varying attachments between pol epsilon and CMG likely regulates DNA unwinding and perhaps explains the previously observed functional synergy between these 2 motors. Aside from a few issues discussed below, I consider the data to be of high quality and of significant novel interest to both the DNA replication field as well as to the broader biological community.

However, there are a few issues in need of improvement.

Larger issues.

1. Potential limitations on the structural model. This manuscript solves a variety of new Cryo-EM structures in the 3-4 angstrom resolution range. Included with this manuscript is the PDB summary for one of their best structures. However, this report indicates that for most of the proteins in the complex, the model lacks a substantial amount the C-terminal sequences (for example, the large subunit of pol epsilon, with a length of 2222 amino acids, is only represented in the structure as the N-terminal most 813 amino acids). As the C-terminal parts of both the CMG complex and pol epsilon are rather heavily featured in this manuscript, this seems like a problem in need of explanation and a possible mention in the manuscript.

2. Functional relevance of CMG and Pol epsilon interactions? This manuscript is entitled "Molecular Mechanism Coordinating the Activities of CMG Helicase and Leading Strand DNA Polymerase at Replication Fork" and implies that the physical interactions between CMG and pol epsilon are of likely major regulatory significance for DNA unwinding. However, no functional data is included to support the structural analysis, and there is no direct evidence that these physical interactions "coordinate" anything. The structural data in this manuscript facilitates the design of an appropriate pol2 mutant that blocks the relevant interactions with CMG. Some biochemistry please! – Does the loss of the key CMG/Pol epsilon interaction destroy the previously observed unwinding synergy? This is really the necessary experiment to nail the main finding in this paper.

Minor issues:

3. In the Methods section, some of the proteins contain epitope tags which were used for their purification. However, the identity of these proteins are shrouded in cited references. In each case, please briefly mention which protein is epitope tagged, the nature of the tag, and its location in the protein.

4. There are several sentences that could use some clarification:
137. Due to the limited resolution of the Pol ϵ -CMG interfaces ($\sim 7.5 \text{ \AA}$), the
138. detailed interactions between Pol ϵ and CMG helicase were not resolved.
In the in the current study or in a prior study?

130. In addition, while the lagging strand DNA is highly dynamic and was not resolved in our model, ...
5. Ambiguous labelling of Figure 6. The panels in this figure contain the descriptors "Pol2 interacting helices, Dbp2 interacting helices, and Pol2-ZF2 containing helices". These descriptors are labelling the key structural transition between complex 1 and V. However, for instance are the "Pol2 interacting helices" part of Pol2, or the part of CMG that interacts with Pol2? Please either improve the legend or find a better way to label the figures.

6. In Figure 7, the DPB3/4 subunits are missing in panels e and f.

Reviewer #2 (Remarks to the Author):

Xu and colleagues report several cryo-EM structures of the *S. cerevisiae* leading strand replisome containing CMG-pole-Ctf4-Tof1-Csm3 at resolutions of 3.2-7.3Å. The configurations of CMG, fork protection components, pole and their associations are very similar to those reported previously for the CMGE at 4.9Å resolution, the yeast fork protection complex at 3.5Å resolution and the human replisome at 3.2Å resolution. In their manuscript, the authors describe (1) four docking sites of pole onto CMG, (2) different conformers for pole engagement of CMG and (3) different ATPase states of the MCM ring in CMG and their relations to leading strand DNA and pole conformers. While the slightly higher resolution structure compared to the human replisome may be useful for aficionados studying the yeast replisome, this work recapitulates and reinforces many conclusions of previous studies and structures without much novel mechanistic insight and may be more suitable for a more specialized journal.

Major concerns:

1) The authors' hypothesis that the non-catalytic Pol2-CTD binding to Mcm2/5 specifically induces stalling is interesting but it has not been directly tested that this specific interface is responsible and not another one. Changes in the C-tier of MCM in CMG have been observed previously and correlated with the mode of pole engagement in the yeast CMGE structure and the human replisome, which led to the proposal that Pol2 engagement modulates CMG helicase activity. Establishing the precise mechanism, however, would require biochemical or biophysical measurements of this coupling and the consequences for both CMG unwinding and DNA synthesis through targeted mutagenesis. Without such experiments the authors' conclusion (which is the only novel aspect of replisome mechanism in this manuscript) is premature.

2) The pull-downs shown in Ext. data fig. 5 are not convincing. The differences between full-length and truncated pole in associating with CMG appear to be minor at best when looking at Pol2 bands. Moreover, other subunits besides Pol2 are not clearly visible in the gels in pull-downs, including Dpb2, which the authors argue mediates pole tethering to CMG in the absence of DNA through the Dpb2-NTD. If Dpb2-NTD were indeed the anchor point, then a Dpb2 protein band should be clearly visible in the pull-down reactions without DNA, but it looks like only Pol2 is attached to CMG after salt washing. This banding pattern also seems to argue against on-off cycling of Pol2 from the MCM ring since similar amounts of Pol2 are recovered with full-length and Dpb2-delta-NTD pole reactions. Negative controls to rule out unspecific binding are also missing.

3) Page 4 line 82: The authors' statement that "the catalytic NTD of Pol2 was removed" in a prior 4.9Å resolution CMGE structure determined by Goswami et al. is incorrect. The sample used in that study was reconstituted with full-length pole that contained mutations in the NTD of Pol2 exonuclease. Due to flexibility, the Pol2-NTD was not visible in this structure.

- 4) Could the author clarify whether the different stabilities of pole cause any alterations in the interface between Psf1 and Dpb2-NTD?
- 5) Fig. 5: It is not explained whether the nucleotides shown in green are ATPgS or ADP.
- 6) Extended data fig. 10: The meaning of the green and blue colours is not explained. It would be less confusing if each subunit would be coloured consistently with the same colour in each panel.
- 7) Page 12, lines 356/357: The work by Eickhoff et al. (Cell Reports 2019) and Rzechorzek et al. (Nucleic Acids Research 2020) should be cited here as well.
- 8) Table 1: The authors use a FSC threshold of 0.143 to report model resolution, which causes overestimation of model resolution. It is common practice to use a FSC threshold of 0.5 when comparing models to cryoEM maps and should be done by the authors as well.
- 9) Ext. data fig. 3: It is not described how the electrostatic potential maps were calculated. A colour code is also not provided for the electrostatic surface potential.
- 10) Ext. data fig. 3: It is surprising that the authors did not use the structure of the human replisome (Jones et al. EMBO Journal 2021) for their comparison of yeast and human Dpb2-Psf1 interactions and instead model it based on separate structures.

Minor comments:

- 1) Page 8 line 203: Fig. 3e doesn't match the description in the main text.
- 2) Page 11 line 304: Fig. 6a should be Fig. 5a.
- 3) Fig. 1: Csm3 and Cdc45 are both coloured green. Using different colours for Csm3 and Cdc45 would improve clarity.
- 4) Page 9 line 253: H2I should be defined.
- 5) Page 18 line 537: "XXX" needs to be clarified.

We thank the reviewers for their constructive comments, which we have taken seriously and addressed one by one as listed below. As a result, the paper is much improved in accessibility and quality.

REVIEWER COMMENTS

Reviewer #1 (Remarks to the Author):

Summary. Replication forks contain three different molecular motors: the helicase (CMG complex), and leading and lagging strand DNA polymerases (epsilon and delta). Although each of these motors have independent activities, these activities synergistically change when the motors are combined on a single DNA substrate. However, a molecular explanation for this synergy is currently lacking. Using purified proteins and a forked DNA substrate, Xu et. al., reconstituted physiologically relevant leading strand complexes containing CMG and Pol epsilon. Through the inclusion of ATPγS, a limited array of conformational isomers was formed that presumably reflect different snap-shots in the leading strand reaction cycle. Using high-resolution Cryo EM, the resulting structural models demonstrated that pol epsilon is attached to the CMG complex at multiple sites. Interestingly, different attachment patterns between pol epsilon and CMG varied among the structures and correlated strongly with DNA attachment to specific CMG subunits and nucleotide occupancy in specific CMG active sites. These correlations implied that the varying attachments between pol epsilon and CMG likely regulates DNA unwinding and perhaps explains the previously observed functional synergy between these 2 motors. Aside from a few issues discussed below, I consider the data to be of high quality and of significant novel interest to both the DNA replication field as well as to the broader biological community.

We thank the reviewer for these valuable comments.

However, there are a few issues in need of improvement.

Larger issues.

1. Potential limitations on the structural model. This manuscript solves a variety of new Cryo-EM structures in the 3-4 angstrom resolution range. Included with this manuscript is the PDB summary for one of their best structures. However, this report indicates that for most of the proteins in the complex, the model lacks a substantial amount the C-terminal sequences (for example, the large subunit of pol epsilon, with a length of 2222 amino acids, is only represented in the structure as the N-terminal most 813 amino acids). As the C-terminal parts of both the CMG complex and pol epsilon are rather heavily featured in this manuscript, this seems like a problem in need of explanation and a possible mention in the manuscript.

We thank the review for the suggestion. The model (PDB: 7VVF) was built mainly against the map (EMD-32140). To further improve map quality of the interested regions, we performed focused refinement (Extended Data Figure 1m). The resultant maps were then used for model building/adjusting of relevant regions. In the final model (PDB: 7VVF), the C-terminal part of Pol2 (1317-2222) are unambiguously built, while its N-terminal part is missing due to the flexibility. For other models representing states II-V (PDB: 7VVQ, 7VXJ, 7VWU, 7VWI), the model of state I was fitted into the corresponding cryo-EM maps, and rigid alignment was performed in Chimera to match with the conformational changes but without adjusting side chains. We also made clear description in the model building of the method section to avoid confusion on page 26.

In addition, it seems that there is a possibility of confusion regarding the graphic illustration of the residue-property plots. Take Mcm2 as an example (see the following figure extracted from the PDB validation report). Its plot only summarizes the residue properties related to different outlier classes. Of note, 24% of the residues colored in grey does not belong to the C-terminal part of Mcm2 but rather indicates the percentage of the residues that are not modeled.

2. Functional relevance of CMG and Pol epsilon interactions? This manuscript is entitled “Molecular Mechanism Coordinating the Activities of CMG Helicase and Leading Strand DNA Polymerase at Replication Fork” and implies that the physical interactions between CMG and pol epsilon are of likely major regulatory significance for DNA unwinding. However, no functional data is included to support the structural analysis, and there is no direct evidence that these physical interactions “coordinate” anything. The structural data in this manuscript facilitates the design of an appropriate pol2 mutant that blocks the relevant interactions with CMG. Some biochemistry please! – Does the loss of the key CMG/Pol epsilon interaction destroy the previously observe unwinding synergy? This is really the necessary experiment to nail the main finding in this paper.

We thank the reviewer for the suggestions. In the revised version, we have included functional data to support the structural analyses (Fig. 2 and 6). Using the structural data as a guide, we generated a series of *pol2* mutants, aiming to weaken or abolish Polε’s binding to the MCM ring (Fig. 2a-b). Our findings demonstrate that the interaction between Pole and the MCM ring plays a crucial role in regulating replication initiation (Fig. 2d-e). When this interaction is interrupted, CMG formation is largely inhibited (Fig. 2e), resulting in a significant delay in S phase entry in the mutant cells. While the coupling between Pole and the MCM ring is dispensable for S phase progression *in vivo* (Fig. 2f), our *in vitro* assays indicate that it is important for leading-strand DNA synthesis mediated by Pole (Fig. 6e-g) while Pole binding to MCM ring does not obviously affect the overall rate of CMG helicase

in DNA unwinding (Fig. 6a-d). Together, our structural and functional analyses highlight the importance of Pol ϵ 's periodic binding to the motor domain of the CMG in helicase activation and leading-strand DNA synthesis.

Figure 2. The physiological function of Pol ϵ -MCM coupling in replication initiation and S phase progression.

(a) *in vitro* CMG pull-down assays to examine the binding affinity of Pol ϵ , Pol ϵ Δ 2N, and the related mutants (Pol2 Δ ADS2 and Pol2 Δ ADS2+4) to CMG in the presence or absence of DNA.

(b) Schematic illustration of pol2 mutant construction.

(c) Pol ϵ -MCM coupling is essential for cell viability. The indicated truncation mutants of Pol2 were tested for complementation of a depletion of Pol2-AID, the expression of which is under the control of Tet promoter. YPD: yeast extract-peptone-dextrose; NAA: 1-naphthalene acetic acid; Dox, doxycycline.

(d) Flow cytometry profiles of cells collected from the α factor block-and-release assay with the indicated pol2-iAID strains under restrictive condition.

(e) The lysates of the indicated samples were prepared for Flag IP with anti-FLAG beads. Each sample was analyzed with antibodies against Flag, Cdc45, and Mcm2.

(f) Flow cytometry profiles of cells collected similar to d but with the relevant cells pre-synchronized at G1 by α factor before further release into hydroxyurea (HU).

Figure 6. The effect of Polε docking to MCM ring on CMG helicase and leading strand DNA synthesis in vitro.

(a) Schematic illustration of unwinding of a 20-bp forked DNA by CMG.

(b-c) Native PAGE of CMG helicase assays performed as in a in the absence (b) or presence (c) of Polε.

(d) Quantification of the results; assays were performed in triplicate.

(e) Scheme of reconstituting leading-strand DNA synthesis by Polε (see details in methods).

(f) Purified factors used for DNA replication assays analyzed by SDS-PAGE with Coomassie staining. The positions of the relevant polypeptides are indicated.

(g) Alkaline agarose gel of reaction products from the replication assays performed as in e.

Minor issues:

3. In the Methods section, some of the proteins contain epitope tags which were used for their purification. However, the identity of these proteins are shrouded in cited references. In each case, please briefly mention which protein is epitope tagged, the nature of the tag, and its location in the protein.

The suggestion was well taken. The relevant information has been included in the Methods.

4. There are several sentences that could use some clarification:

137. Due to the limited resolution of the Polε-CMG interfaces (~7.5 Å), the
 138. detailed interactions between Polε and CMG helicase were not resolved.
 In the current study or in a prior study?

We thank the reviewer for pointing out this issue. It is in the previous study that the detailed interaction between Polε and CMG were not fully resolved. The relevant text has been removed to avoid confusion.

130. In addition, while the lagging strand DNA is highly dynamic and was not resolved in our model, ...

The suggestion was well taken.

5. Ambiguous labelling of Figure 6. The panels in this figure contain the descriptors “Pol2 interacting helices, Dbp2 interacting helices, and Pol2-ZF2 containing helices”. These descriptors are labelling the key structural transition between complex 1 and V. However, for instance are the “Pol2 interacting helices” part of Pol2, or the part of CMG that interacts with Pol2? Please either improve the legend or find a better way to label the figures.

The suggestion was well taken. We have updated the labelling of the **Figure 5** (the original Figure 6).

Figure 5. The Conformational change in the MCM motor domains regulates Polε docking site formation.

Structural comparison of replisome in State I and State V, showing large displacement in relevant Pol2 (a), Dpb2-OB (b), and Pol2-ZF2 (c) contacting α helices from Mcm2 (a), Mcm3:5 interface (b) and Mcm5-CTD (c).

6. In Figure 7, the DPB3/4 subunits are missing in panels e and f.

We thank the reviewer’s suggestion. **Figure 7** has been updated accordingly.

Figure 7. Model illustrating potential functions of Polε cycling on and off MCM ring.

Reviewer #2 (Remarks to the Author):

We would like to express our sincere gratitude to the reviewer for his/her valuable and constructive comments. The reviewers' input has been taken seriously, and we have made the necessary revisions to address the concerns. We believe that the final version of our manuscript is much improved as a result of the reviewer's contribution.

Xu and colleagues report several cryo-EM structures of the *S. cerevisiae* leading strand replisome containing CMG-pole-Ctf4-Tof1-Csm3 at resolutions of 3.2-7.3Å. The configurations of CMG, fork protection components, pole and their associations are very similar to those reported previously for the CMGE at 4.9Å resolution, the yeast fork protection complex at 3.5Å resolution and the human replisome at 3.2Å resolution. In their manuscript, the authors describe (1) four docking sites of pole onto CMG, (2) different conformers for pole engagement of CMG and (3) different ATPase states of the MCM ring in CMG and their relations to leading strand DNA and pole conformers. While the slightly higher resolution structure compared to the human replisome may be useful for aficionados studying the yeast replisome, this work recapitulates and reinforces many conclusions of previous studies and structures without much novel mechanistic insight and may be more suitable for a more specialized journal.

We agree with the reviewer on the fact that the engagement between Polε and CMG helicase was indeed observed in previous studies with yeast CMGE (Goswami et al 2018) and human replisome (Jones et al 2021). However, it is important to note that in the previous yeast CMGE study, only one static conformation was identified with limited resolution (Goswami et al 2018). The previous human replisome study reported two conformations of Polε: one similar to the low-resolution yeast CMGE and the other one showing a flexible Polε but with an empty DNA binding central channel (Jones et al 2021). All these studies did not provide clear information on the relationship between Polε and CMG helicase in a translocating replisome at the replication fork.

We would like to clarify that the replisome conformations, ranging from States II to V (Fig. 3 and 4), were not reported in the previous studies. Our study represents the first visualization of the dynamic binding of DNA polymerase to the motor domains of CMG helicase to coordinate DNA translocation and leading strand synthesis in the replisome. In the revised version, we also included both *in vivo* and *in vitro* data to establish the functional significance of the Polε-MCM coupling in regulating DNA replication. The new data indicates that the binding of Polε to the MCM CTD ring is required to drive CMG formation during origin firing (Fig. 2). Moreover, the Polε-MCM coupling plays a crucial role in leading-strand DNA synthesis mediated by Polε (Fig. 6).

We believe that the updated version of our work contributes novel mechanistic insights into the inner working of the replisome.

Major concerns:

1) The authors' hypothesis that the non-catalytic Pol2-CTD binding to Mcm2/5 specifically induces stalling is interesting but it has not been directly tested that this specific interface is responsible and not another one. Changes in the C-tier of MCM in CMG have been observed previously and correlated with the mode of pole engagement in the yeast CMGE structure and the human replisome, which led to the proposal that Pol2 engagement modulates CMG

helicase activity. Establishing the precise mechanism, however, would require biochemical or biophysical measurements of this coupling and the consequences for both CMG unwinding and DNA synthesis through targeted mutagenesis. Without such experiments the authors' conclusion (which is the only novel aspect of replisome mechanism in this manuscript) is premature.

We thank the reviewer for the suggestion. We have revised our work to include the results from *in vivo* and *in vitro* assays, which we conducted to better understand the coupling between Pol ϵ and MCM ring. Our *in vivo* experiments indicate that a tight Pol ϵ -MCM coupling is required for replication initiation through driving CMG formation (Fig. 2d-e). In addition, our *in vitro* helicase assays revealed that Pol ϵ 's binding to the MCM-CTD ring does not significantly alter CMG helicase activity (Fig. 6a-d). However, this coupling is required for leading-strand DNA synthesis mediated by Pol ϵ , as shown in our *in vitro* replication assays (Fig. 6e-g).

Figure 2. The physiological function of Pol ϵ -MCM coupling in replication initiation and S phase progression.

(a) *in vitro* CMG pull-down assays to examine the binding affinity of Pol ϵ , Pol ϵ Δ 2N, and the related mutants (Pol2 Δ ADS2 and Pol2 Δ ADS2+4) to CMG in the presence or absence of DNA.

(b) Schematic illustration of pol2 mutant construction.

(c) Pol ϵ -MCM coupling is essential for cell viability. The indicated truncation mutants of Pol2 were tested for complementation of a depletion of Pol2-AID, the expression of which is under the control of

Tet promoter. YPD: yeast extract-peptone-dextrose; NAA: 1-naphthalene acetic acid; Dox, doxycycline.

(d) Flow cytometry profiles of cells collected from the α factor block-and-release assay with the indicated pol2-iAID strains under restrictive condition.

(e) The lysates of the indicated samples from d were prepared for Flag IP with anti-FLAG beads. Each sample was analyzed with antibodies against Flag, Cdc45, and Mcm6.

(f) Flow cytometry profiles of cells collected similar as d but with the relevant cells pre-synchronized at G1 by α factor before further release into hydroxyurea (HU).

Figure 6. The effect of Polε docking to MCM ring on CMG helicase and leading strand DNA synthesis in vitro.

(a) Schematic illustration of unwinding of a 20-bp forked DNA by CMG.

(b-c) Native PAGE of CMG helicase assays performed as in a in the absence (b) or presence (c) of Polε.

(d) Quantification of the results; assays were performed in triplicate.

(e) Scheme of reconstituting leading-strand DNA synthesis by Polε (see details in methods).

(f) Purified factors used for DNA replication assays analyzed by SDS-PAGE with Coomassie staining. The positions of the relevant polypeptides are indicated.

(g) Alkaline agarose gel of reaction products from the replication assays performed as in e.

2) The pull-downs shown in Ext. data fig. 5 are not convincing. The differences between full-length and truncated pole in associating with CMG appear to be minor at best when looking at Pol2 bands. Moreover, other subunits besides Pol2 are not clearly visible in the gels in pull-downs, including Dpb2, which the authors argue mediates pole tethering to CMG in the absence of DNA through the Dpb2-NTD. If Dpb2-NTD were indeed the anchor point, then a Dpb2 protein band should be clearly visible in the pull-down reactions without DNA, but it looks like only Pol2 is attached to CMG after salt washing. This banding pattern also seems to argue against on-off cycling of Pol2 from the MCM ring since similar amounts of Pol2 are recovered with full-length and Dpb2-delta-NTD pole reactions. Negative controls to rule out unspecific binding are also missing.

We thank the reviewer for pointing out these issues. We have repeated the pull-down assays using Polε (WT and mutants) as the bait to co-precipitate CMG. Our results show a

significant difference between the full-length and truncated Pol ϵ (Dpb2 Δ N) in binding to CMG. It is evident that NTD removal from Dpb2 leads to an obvious reduction in the affinity of Pol ϵ associating with CMG in the presence of DNA (Fig. 2a, lanes 6-7). This result suggests that the MCM ring could mediate a direct binding of Pol ϵ to CMG helicase, independent of Dpb2-NTD. When DNA is absent, the Pol ϵ Δ 2N-MCM coupling is largely abolished while Pol ϵ -WT can still maintain a strong interaction with CMG (Fig. 2a, lanes 10-11). Together, these results support the conclusion that DNA translocation by CMG mediates the on-off cycling of Pol2 from the MCM ring, which uses the interaction between Dpb2-NTD and Psf1 as a hinge. This pull-down data is added as Fig. 2a in the revised manuscript.

Figure 2. The physiological function of Pol ϵ -MCM coupling in replication initiation and S phase progression.

(a) *in vitro* CMG pull-down assays to examine the binding affinity of Pol ϵ , Pol ϵ Δ 2N, and the related mutants (Pol2 Δ DS2 and Pol2 Δ DS2+4) to CMG in the presence or absence of DNA.

(b) Schematic illustration of pol2 mutant construction.

3) Page 4 line 82: The authors' statement that "the catalytic NTD of Pol2 was removed" in a prior 4.9Å resolution CMGE structure determined by Goswami et al. is incorrect. The sample used in that study was reconstituted with full-length pole that contained mutations in the NTD of Pol2 exonuclease. Due to flexibility, the Pol2-NTD was not visible in this structure.

We thank the reviewer for pointing out this mistake. It has been fixed on page 4.

4) Could the author clarify whether the different stabilities of pole cause any alterations in the interface between Psf1 and Dpb2-NTD?

To understand the potential impact of different stabilities of Pol ϵ on the Psf1-Dpb2-NTD interface, we conducted a comparison of the replisome structures in state I and state V (Fig. R1 a and b). Our result shows a slight shift in the helices α 1-4 of Dpb2-NTD (Fig. R1c), leading to the helices α 1-2 moving toward Psf1. It is likely that the interaction between Psf1

and Dpb2 may be strengthened as a result of this alteration. However, due to the limited resolution of the state V replisome, the precise effects of this conformational change on the coupling between Psf1 and Dpb2-NTD could not be fully unraveled at this stage.

Figure R1. The interface between Psf1 and Dpb2-NTD. (a and b) The conformations of Polε in the replisomes, state I (a) and state V (b). (c) Superimposition of the Psf1-Dpb2-NTD interfaces from state I (Psf1 in cyan and Dpb2 in magenta) and state V (grey) using Psf1 as a reference.

5) Fig. 5: It is not explained whether the nucleotides shown in green are ATPγS or ADP.

In the original Fig. 5, the nucleotide identity (ATPγS or ADP) can only be determined in the state I replisome as shown in Extended Data Figure 9. However, the limited resolution of the maps for other states (II-V) makes it difficult to discern nucleotides with certainty. We have updated the figure to focus solely on DNA translocation.

6) Extended data fig. 10: The meaning of the green and blue colours is not explained. It would be less confusing if each subunit would be coloured consistently with the same colour in each panel.

The suggestion was well taken. The colour scheme has been updated to avoid confusion (see Extended Data Fig. 5).

Extended Data Figure 5. DNA translocation driven by displacement of ATPase loops around the motor domains of the MCM ring.

(a-b) Superimposition of the MCM-CTD rings from State I and State V using Mcm3 as a reference, highlighting the orientation change of the MCM motor domain.

(c-h) Comparison of H2I and PS1 loops of each inter-subunit interface from State I (blue) and State V (cyan) showing movement of ATPase loops for DNA threading. Loops of the left subunit used as a reference for superimposition. For clarity, DNA from State V replisome is not shown in c-e; and DNA from State I not shown in f-h. The directions and distances of PS1 displacements are marked with red arrows.

7) Page 12, lines 356/357: The work by Eickhoff et al. (Cell Reports 2019) and Rzechorzek et al. (Nucleic Acids Research 2020) should be cited here as well.

We appreciate the reviewer's comment. The citation problem has been fixed.

8) Table 1: The authors use a FSC threshold of 0.143 to report model resolution, which causes overestimation of model resolution. It is common practice to use a FSC threshold of 0.5 when comparing models to cryoEM maps and should be done by the authors as well.

The suggestion was well taken. The FSC threshold of 0.5 and corresponding resolution was updated (Table 1).

9) Ext. data fig. 3: It is not described how the electrostatic potential maps were calculated. A colour code is also not provided for the electrostatic surface potential.

We thank the reviewer for pointing out this issue. The relevant information related to the electrostatic potential map was included in the legend of **Extended Data Fig. 3**. The colour code was also provided in **Extended Data Fig. 3**.

Extended Data Figure 3. Electrostatic analysis of Dpb2-Psf1 interface.

(a) Multiple sequence alignment of Dpb2-NTD from various species as indicated. The locations of α helices are labelled at top.
 (b-d) Interaction between Psf1 B domain and Dpb2-NTD. The electrostatic surface potential maps of these two domains, calculated are displayed in **b** and **c** respectively. **c** is same as **b** but with indicated rotation, highlighting a polarized pattern of the surface charge on Dpb2-NTD.
 (e) hPOLE2-NTD from a human replisome structure (PDB: 7PFO) is superimposed with ScDpb2-NTD from our yeast replisome structure.
 (f-h) same as **a-c** but shown with hPOLE2-NTD and hPsf1 B domain (PDB: 7PFO). The electrostatic potentials were calculated with the APBS plugin in ChimeraX using the default settings and displayed with a range of ± 10 kT/e.

10) Ext. data fig. 3: It is surprising that the authors did not use the structure of the human replisome (Jones et al. EMBO Journal 2021) for their comparison of yeast and human Dpb2-Psf1 interactions and instead model it based on separate structures.

We appreciate the reviewer’s comment. In the revised version, the comparison of Dpb2-Psf1 interaction was performed with the structure of the human replisome (PDB: 7PFO) (Jones et al 2021 EMBO J).

Minor comments:

1) Page 8 line 203: Fig. 3e doesn’t match the description in the main text.

We thank the reviewer for pointing out this mistake. It has been fixed.

2) Page 11 line 304: Fig. 6a should be Fig. 5a.

We thank the reviewer for pointing out this issue. We have corrected it.

3) Fig. 1: Csm3 and Cdc45 are both coloured green. Using different colours for Csm3 and Cdc45 would improve clarity.

The suggestion was well taken. The colour of Csm3 has been changed to purple (Extended Data Fig. 2).

Extended Data Figure 2. Overall structure of a leading-strand replisome at replication fork.

4) Page 9 line 253: H2I should be defined.

The suggestion was well taken.

5) Page 18 line 537: “XXX” needs to be clarified.

We are sorry for this mistake. It has been fixed.

REVIEWER COMMENTS

Reviewer #1 (Remarks to the Author):

Nice job with the revised manuscript!

Reviewer #2 (Remarks to the Author):

In this revision, Xu et al. include new biochemical data that improve the manuscript and their main conclusions. However, they also raise some issues that should be addressed.

The results of the helicase assays shown in the manuscript lack important controls to convince the reader that the activity measured is translocation-dependent CMG helicase activity rather than destabilization of the short duplex DNA (20 bp) by CMG binding due to the large excess of CMG in their assays, or that it is caused by the unwinding activity of a protein contaminant. Is unwinding activity lost when ATPgammaS is used? Does a helicase-dead CMG mutant abolish the helicase activity in the authors' setup? Have the authors tried a longer duplex to see coupling of Pol epsilon and CMG in helicase assays? Additionally, the authors should comment on why they didn't include a capture oligo in their helicase assays. Such oligos are typically used to prevent reannealing of the unwound strands.

The in vitro replication assays show that DNA synthesis correlates with the affinity of Pol epsilon for CMG binding. This supports that tethering of the polymerase to the helicase is important as expected but does not directly show that Pol epsilon cycling on and off the MCM ring during CMG translocation as seen in the presented structures is functionally relevant. What happens if Pol epsilon is prevented from cycling off, for example by strengthening the binding interface or by introducing crosslinks? If the cycling is important, one would expect to see a decrease in DNA synthesis in replication assays.

REVIEWER COMMENTS

Reviewer #2 (Remarks to the Author):

In this revision, Xu et al. include new biochemical data that improve the manuscript and their main conclusions. However, they also raise some issues that should be addressed.

The results of the helicase assays shown in the manuscript lack important controls to convince the reader that the activity measured is translocation-dependent CMG helicase activity rather than destabilization of the short duplex DNA (20 bp) by CMG binding due to the large excess of CMG in their assays, or that it is caused by the unwinding activity of a protein contaminant. Is unwinding activity lost when ATP γ S is used? Does a helicase-dead CMG mutant abolish the helicase activity in the authors' setup?

We appreciate the reviewer's valuable suggestions. Regarding the concern about the potential of CMG to destabilize short duplex DNA, we would like to point out that a previous study conducted by Michael O'Donnell's group (Yao et al 2022 PNAS) has addressed this issue. According to the findings of this study, when AMPPNP (Yao et al 2022, Fig. 4B) or a helicase-dead mutant (Yao et al 2022, Fig. 3A) was used, the helicase activity of CMG is largely abolished, indicating that the observed unwinding activity is translocation-dependent CMG helicase activity.

[redacted]

(Yao et al 2022, Fig. 4B)

[redacted]

(Yao et al 2022, Fig. 3A)

Have the authors tried a longer duplex to see coupling of Pol epsilon and CMG in helicase assays?

In response to the reviewer's suggestion, we have included the data from the *in vitro* helicase assays with a longer duplex DNA (60 bp) to assess the impact of Pol ϵ -MCM coupling. Our results show a minor suppression of CMG's helicase activity by Pol ϵ (Extended Data Fig. 10), consistent with the results obtained with the shorter duplex DNA (20 bp) (Fig. 6a-d).

Extended Figure 10. The effect of Pol ϵ on CMG helicase activity in vitro.

(a) Schematic illustration of unwinding of a 60-bp forked DNA by CMG. (b-c) Native PAGE of CMG helicase assays performed as in a in the absence (b) or presence (c) of Pol ϵ . (d) Quantification of the results.

Additionally, the authors should comment on why they didn't include a capture oligo in their helicase assays. Such oligos are typically used to prevent reannealing of the unwound strands.

We adopted our protocol for assaying helicase activities with 20-bp forked DNA from Yao et al (2022, PNAS), where a trap oligo was not included. However, when we performed assays with 60-bp forked DNA, we used a capture oligo to prevent reannealing of the unwound strands (see the method on page 27).

The in vitro replication assays show that DNA synthesis correlates with the affinity of Pol epsilon for CMG binding. This supports that tethering of the polymerase to the helicase is important as expected but does not directly show that Pol epsilon cycling on and off the MCM ring during CMG translocation as seen in the presented structures is functionally relevant. What happens if Pol epsilon is prevented from cycling off, for example by strengthening the binding interface or by introducing crosslinks? If the cycling is important, one would expect to see a decrease in DNA synthesis in replication assays.

We appreciate the insightful comments from the reviewer. In this manuscript, our findings highlight the significance of the Pol ϵ -MCM coupling in replication initiation and leading strand DNA synthesis. Further investigations are required to further elucidate the consequence(s) of

continuous tethering of Pol ϵ to the MCM ring during CMG translocation. Considering the potential roles of Pol ϵ in parental histone recycling, we anticipate that disrupting the periodic binding of Pol ϵ to the MCM ring through enforced coupling may impede the normal transfer of parental histones to newly synthesized leading strand. We have included this intriguing point in the discussion on page 16.

REVIEWERS' COMMENTS

Reviewer #2 (Remarks to the Author):

The authors have addressed all remaining concerns.